# Characterisation of the biflavonoid hinokiflavone as a pre-mRNA splicing modulator that inhibits SENP

Andrea Pawellek[1], Ursula Ryder[1], Triin Tammsalu[1], Lewis J King[2], Helmi Kreinin[2], Tony Ly[1], Ronald T Hay[1], Richard C Hartley[2], Angus I Lamond[1]*

[1]Centre for Gene Regulation and Expression, School of Life Sciences, University of Dundee, Dundee, United Kingdom; [2]WestCHEM, School of Chemistry, University of Glasgow, Glasgow, United Kingdom

**Abstract** We have identified the plant biflavonoid hinokiflavone as an inhibitor of splicing in vitro and modulator of alternative splicing in cells. Chemical synthesis confirms hinokiflavone is the active molecule. Hinokiflavone inhibits splicing in vitro by blocking spliceosome assembly, preventing formation of the B complex. Cells treated with hinokiflavone show altered subnuclear organization specifically of splicing factors required for A complex formation, which relocalize together with SUMO1 and SUMO2 into enlarged nuclear speckles containing polyadenylated RNA. Hinokiflavone increases protein SUMOylation levels, both in in vitro splicing reactions and in cells. Hinokiflavone also inhibited a purified, *E. coli* expressed SUMO protease, SENP1, in vitro, indicating the increase in SUMOylated proteins results primarily from inhibition of de-SUMOylation. Using a quantitative proteomics assay we identified many SUMO2 sites whose levels increased in cells following hinokiflavone treatment, with the major targets including six proteins that are components of the U2 snRNP and required for A complex formation.

DOI: https://doi.org/10.7554/eLife.27402.001

*For correspondence:
a.i.lamond@dundee.ac.uk

Competing interests: The authors declare that no competing interests exist.

## Introduction

Pre-mRNA splicing is an essential step in gene expression in eukaryotes. During splicing, intron sequences are removed from nascent, pre-mRNA gene transcripts via two, sequential transesterification reactions, thereby joining exon sequences to generate messenger RNA (mRNA) for protein translation (reviewed in [*Wahl et al., 2009*; *Papasaikas and Valcárcel, 2016*; *Lee and Rio, 2015*]). The splicing of pre-mRNA takes place in the cell nucleus and is catalyzed by the large (>3 MDa), ribonucleoprotein spliceosome complex. Spliceosome complexes assemble at the splice sites in a pre-mRNA transcript, involving a stepwise assembly pathway of the U1, U2 and U4/5/6 snRNP spliceosome subunits, together with additional protein splicing factors. The core splicing machinery, spliceosome assembly pathway and reaction mechanism is highly conserved across eukaryotes. Protein splicing factors have been shown to be targets for post translational modifications, including acetylation, phosphorylation and SUMOylation, which can affect the efficiency of spliceosome assembly and splicing (*Chen and Moore, 2014*; *Pozzi et al., 2017*). In mammalian cells the splicing machinery typically shows a punctate, or 'speckled' localisation pattern in the nucleus, with snRNPs also located in bright nuclear foci, which are termed Cajal bodies (*Lamond and Spector, 2003*).

In higher eukaryotes, most pre-mRNA transcripts have multiple introns and it is common for a single gene to give rise to multiple different mRNA products through alternative patterns of intron removal, e.g. via either the choice of different 5' and/or 3' splice sites, or via exon skipping etc. (reviewed in [*Lee and Rio, 2015*; *Chabot, 2015*]). Alternative splicing is thus a major mechanism for generating proteome diversity in higher eukaryotes and is likely to have played a major role in the

evolution of organismal complexity. Changes in the pattern of alternative splicing of pre-mRNAs is highly regulated and plays an important role during organismal development, cellular differentiation and the response to many physiological stimuli. For example, a number of genes regulating apoptosis pathways, such as BCL2 and MCL2, can be alternatively spliced to generate distinct mRNAs encoding proteins that either promote, or inhibit, cell survival (*Wu and Tang, 2016*; *David and Manley, 2010*).

Recent high-throughput, deep sequencing experiments have shown that almost all human genes produce alternatively spliced mRNA isoforms (*Pan et al., 2008*). The dysregulation of alternative splicing is also now recognized as a major mechanism for multiple forms of human disease, including inherited disorders, cancer, diabetes and neurodegenerative diseases (*Le et al., 2015*). The ability to target the modulation of specific classes of alternative pre-mRNA splicing events using small molecules is thus seen as an important new therapeutic strategy for drug development and future disease therapy (*Woll et al., 2016*; *Ratni et al., 2016*; *Naryshkin et al., 2014*).

Chemical compounds are widely used as biotools to study the gene expression machineries involved in transcription and translation. In contrast, well-characterized compounds that can be used to dissect and study the splicing machinery are still very limited. Most splicing inhibitors described to date are natural products. The best-studied group are all SF3B1 inhibitors, which have either been isolated from the broth of bacteria, or are their synthetic derivatives, like Spliceostatin A, E7107 and herboxidine (*Kaida et al., 2007*; *Kotake et al., 2007*; *Webb et al., 2013*). In addition to the SF3B1 inhibitors, another plant derived compound, the biflavone isoginkgetin (extracted from the leaves of the ginko tree), has been reported to inhibit splicing, but its target remains unknown (*O'Brien et al., 2008*).

Biflavones such as isoginkgetin belong to a subclass of the plant flavonoid family, which have been reported to possess a broad spectrum of pharmacological activity, including antibacterial, anti-cancer, antiviral and anti-inflammatory functions (*Kumar and Pandey, 2013*). In view of their broad biological activity, we tested a set of biflavones for their ability to change pre-mRNA splicing in vitro and *in cellulo*. This identified the biflavone hinokiflavone as a new general splicing modulator active both in vitro and *in cellulo*. Further analysis indicated that hinokiflavone promotes unique changes in subnuclear structure and dramatically increases SUMOylation of a subset of spliceosome proteins by inhibiting SENP1 protease activity.

## Results

### Biflavonoids as splicing modulators

We tested a set of bioflavonoids for a potential effect on pre-mRNA splicing in vitro, including amentoflavone, cupressuflavone, hinokiflavone and sciadopitysin (*Figure 1A*). As a positive control we included the previously reported biflavonoid splicing inhibitor, isoginkgetin (*O'Brien et al., 2008*). Initially, we screened each compound at a high final concentration (500 µM), for a potential inhibitory effect on splicing of the model Ad1 and HPV18 E6 pre-mRNAs, using HeLa nuclear extract in conjunction with a non-radioactive RT-PCR in vitro splicing assay (see Materials and methods). Cupressuflavone did not inhibit splicing under these conditions, while hinokiflavone, amentoflavone, isoginkgetin and sciadopitysin all showed some inhibition of in vitro splicing of the adenovirus and/ or the HPV18E6 pre-mRNAs. The strongest inhibitory effects were obtained with hinokiflavone and amentoflavone, which had not previously been identified as affecting either splicing, or spliceosome assembly (*Figure 1B*). We note that the active compounds each have a linkage connecting the B and A' units of the two flavone moieties.

Next, we tested the biflavonoids for their ability to alter pre-mRNA splicing in human cell lines. First, HEK293 cells were treated for 24 hr with 20–100 µM of each compound and a RT-PCR assay was used to detect potential changes in the splicing patterns of HSP40, RIOK3, RBM5, FAS and MCL1 pre-mRNAs. With the set of pre-mRNAs tested, amentoflavone, cupressuflavone and sciadopitysin did not alter splicing of the tested transcripts (*Figure 2—figure supplement 1*), whereas isoginkgetin and hinokiflavone induced changes in alternative splicing patterns *in cellulo*.

Hinokiflavone showed a stronger effect on pre-mRNA splicing than isoginkgetin, both in vitro and *in cellulo*. Results for isoginkgetin treated HeLa and HEK293 cells are shown in *Figure 2—figure supplement 2*. HeLa, HEK293 and NB4 cells were treated with either DMSO (control), or with

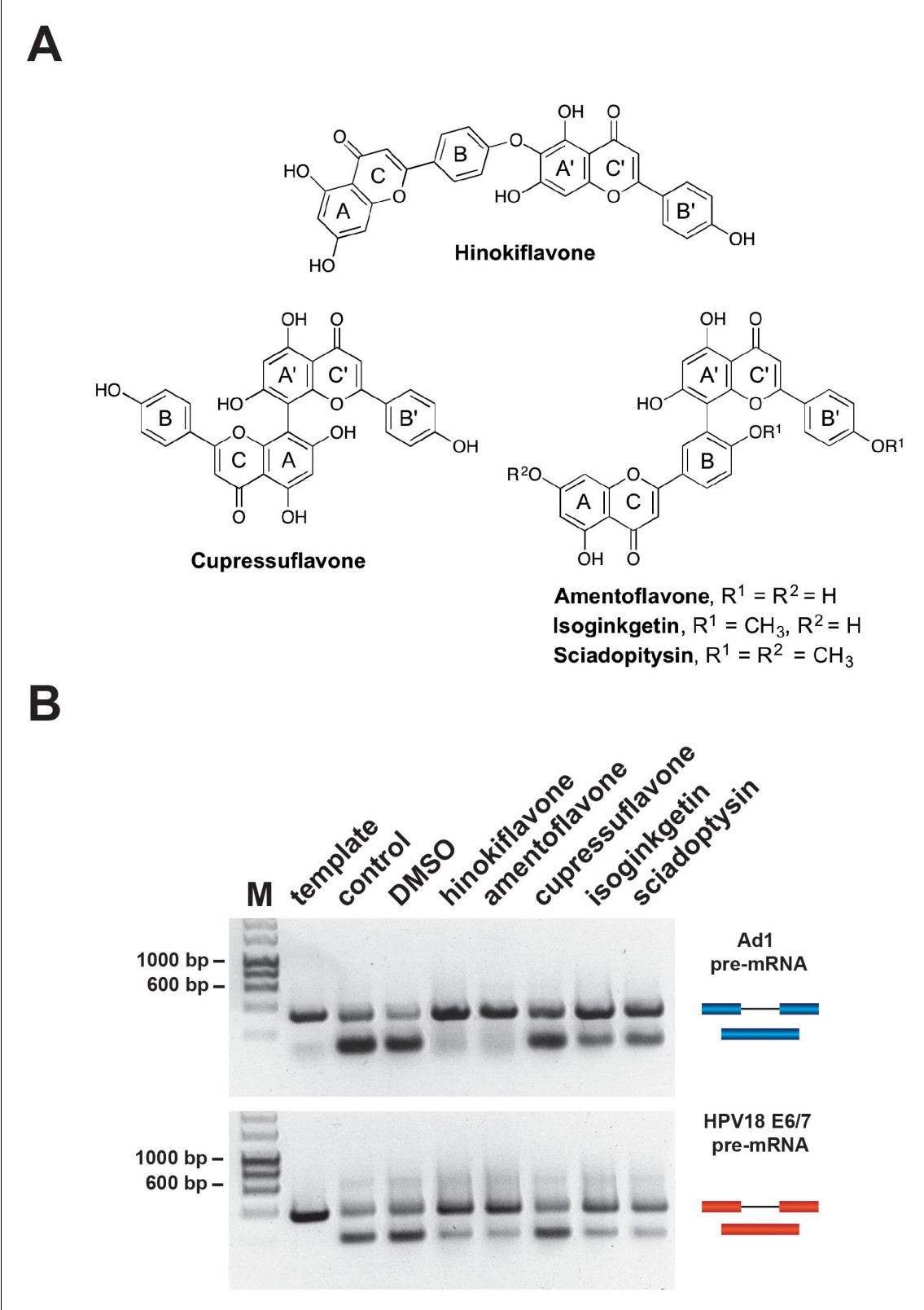

**Figure 1.** Biflavones inhibit splicing in vitro. (**A**) Chemical structures of the five biflavones: hinokiflavone, amentoflavone, cupressuflavone, isoginkgetin and sciadopitysin. These were each tested for their ability to inhibit splicing of the Ad1 and the HPV18 E6/E7 pre-mRNAs in a nonradioactive RT-PCR based in vitro splicing assay. (**B**) Splicing assays show that all compounds except cupressuflavone inhibited splicing to varying degrees in this assay system.

*Figure 1 continued on next page*

*Figure 1 continued*

DOI: https://doi.org/10.7554/eLife.27402.002

hinokiflavone at 10 µM, 20 µM, or 30 µM for 24 hr, then cells were harvested and total RNA extracted. A semiquantitative RT-PCR assay was performed, targeting a representative set of previously studied pre-mRNA transcripts. Thus, we used primer pairs to detect either changes in alternative pre-mRNA splicing for transcripts from the MCL1, NOP56, EIF4A2 and FAS genes, or changes in intron retention for transcripts from the HSP40, RIOK3, ACTR1b and DXO genes. Hinokiflavone modulated splicing in all three cell lines tested, but with the greatest effect observed in NB4 cells (*Figure 2*).

An interesting differential effect on the alternative splicing of MCL1 was observed between the cell lines. Thus, in HeLa and HEK293 cells, hinokiflavone promoted exon 2 skipping, thereby increasing the proportion of the pro-apoptotic mRNA isoform, whereas, in NB4 cells, multiple, aberrant alternatively spliced MCL1 isoforms were detected. In the presence of 10 µM hinokiflavone, a third band of around 1.5 kb was detected next to the expected MCL1-S (pro-apoptotic) and MCL1-L (anti-apoptotic) mRNA isoforms. In addition to the 1.5 kb band, a loss of the MCL1-S isoform and the appearance of a larger splice variant migrating between MCL1-S and MCL1-L was observed in cells treated with either 20 µM, or 30 µM hinokiflavone.

Intron retention in the HSP40 and RIOK3 transcripts was mainly detected in HeLa cells at 30 µM, in HEK293 cells at both 20 µM and 30 µM and in NB4 cells at all concentrations of hinokiflavone tested. In the case of ACTR1B, no changes in alternative splicing were detected in HeLa cells, whereas in HEK293 cells, intron inclusion was only observed in the presence of 10 µM hinokiflavone, while in NB4 cells intron inclusion was observed at all concentrations tested. Hinokiflavone also promoted intron 3 and intron 4 retention in the DXO transcripts and exon 4 skipping in the EIF4A2 transcripts in all cell lines and at all concentration tested. In the case of NOP56, hinokiflavone not only induced an increase in the production of NOP56 transcripts, with exon 8 included, but also induced the usage of alternative 5' and 3' splice sites. In the case of FAS, multiple PCR products were detected, which were individually cloned and sequenced (*Figure 2—figure supplement 3*). The FAS isoforms identified were either lacking exon 5, 6, 7 or 8, exon 6 and exon 7, or exons 6, 7 and 8.

To ensure that these observed effects on splicing in vitro and *in cellulo* were indeed caused by hinokiflavone, rather than by some minor product in the commercially available hinokiflavone isolated from a natural source, we developed a synthetic route for generating the hinokiflavone molecule. A detailed description of the synthetic route will be published separately (King et al., unpublished). Importantly, we find that chemically synthesized hinokiflavone is spectroscopically identical to hinokiflavone isolated from a natural source. The synthetic hinokiflavone also caused a similar alteration in the alternative pre-mRNA splicing pattern of MCL1 as observed for hinokiflavone isolated ex vivo (*Figure 2—figure supplement 4*). We conclude that hinokiflavone is therefore the active molecule and is able to modulate pre-mRNA splicing activity.

## Hinokiflavone prevents assembly of the spliceosome B complex

To investigate whether hinokiflavone inhibits splicing by preventing spliceosome assembly, in vitro splicing reactions were carried out using radioactive Ad1 pre-mRNA and either DMSO (control), or 500 µM hinokiflavone. The reactions were analyzed both by denaturing PAGE to detect reaction products and by native gel electrophoresis to monitor spliceosome assembly (*Figure 3*). Hinokiflavone inhibited the formation of both splicing products and intermediates, with no inhibition seen with the DMSO control, in comparison with untreated nuclear extract (*Figure 3A*). After 1 hr incubation, analysis using native gels showed the typical pattern of A, B and C spliceosome complexes in the DMSO control, similar to untreated nuclear extract. However, in the hinokiflavone treated extract, only H/E and A complexes were detected (*Figure 3B*). This indicates that the inhibition of splicing caused by hinokiflavone results from a failure to assemble the B complex during spliceosome assembly. This may either result from a defect in the mechanism required for transition from the A to B complexes, or because a defective 'A-like' complex is formed that cannot be converted to a B complex.

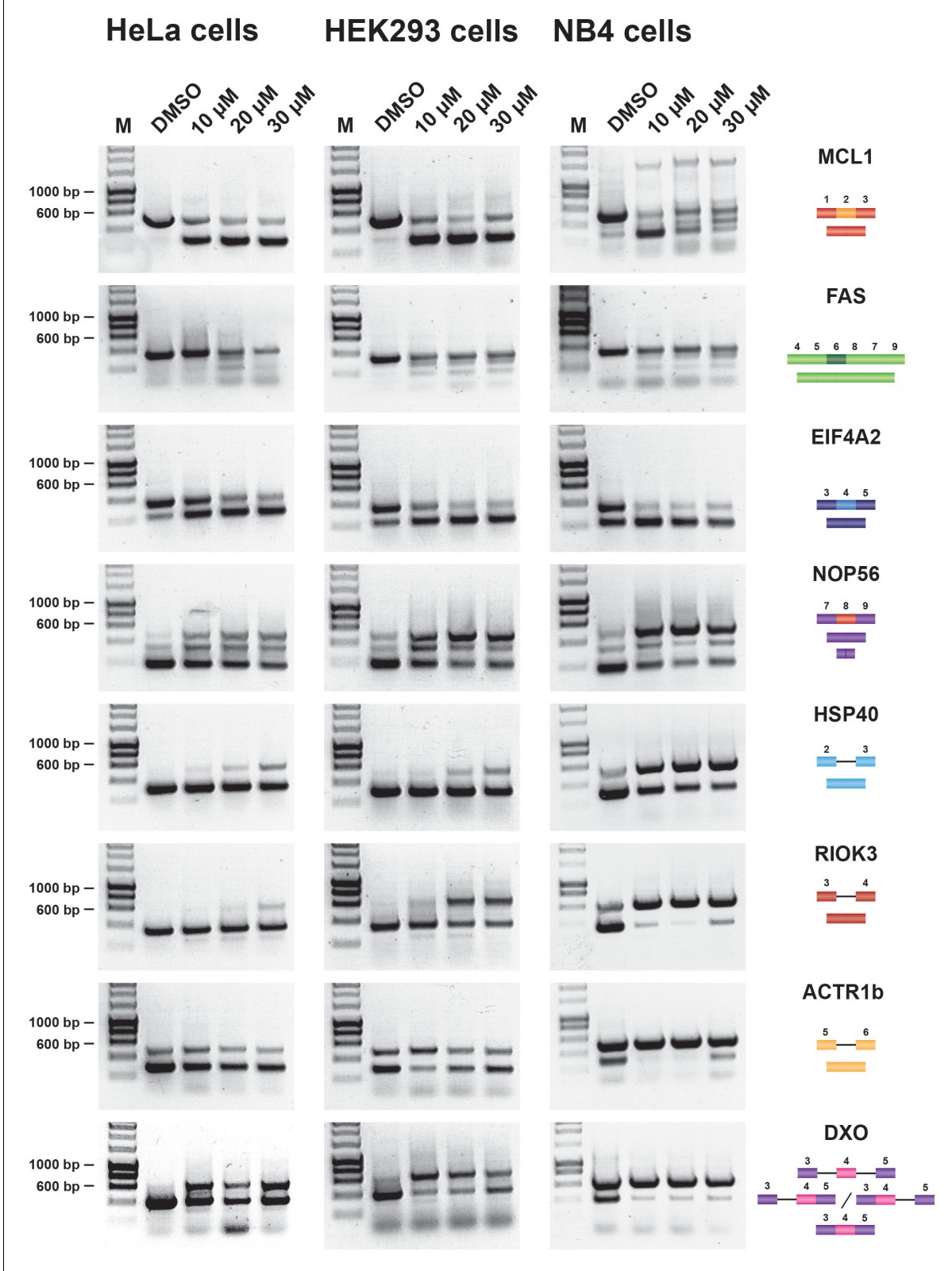

**Figure 2.** Hinokiflavone modulates splicing in cells. Semiquantitative RT-PCR analysis of cells treated with increasing concentrations of either hinokiflavone, or DMSO, for 24 hr. HeLa, HEK293 and NB4 cells were examined for intron inclusion of HSP40, RIOK3, ACTR1b and DXO pre-mRNAs and for exon skipping of MCL1, NOP56, EIF4A2 and FAS pre-mRNAs. The positions of different cDNA products are pictured on the right of the gel images, and the molecular weight markers are shown on the left. Lane M, marker (hyperladder, 1 kb).

*Figure 2 continued on next page*

*Figure 2 continued*

DOI: https://doi.org/10.7554/eLife.27402.003

The following figure supplements are available for figure 2:

**Figure supplement 1.** Amentoflavone, Cupressoflavone and Sciadopitysin do not alter pre-mRNA splicing *in cellulo*.

DOI: https://doi.org/10.7554/eLife.27402.004

**Figure supplement 2.** Isoginkgetin induces splicing changes in HeLa and HEK293 cells.

DOI: https://doi.org/10.7554/eLife.27402.005

**Figure supplement 3.** FAS pre-mRNA splicing in the presence of hinokiflavone.

DOI: https://doi.org/10.7554/eLife.27402.006

**Figure supplement 4.** Comparison of natural and synthetic hinokiflavone.

DOI: https://doi.org/10.7554/eLife.27402.007

## Hinokiflavone blocks cell cycle progression

Next, we tested the effect of hinokiflavone on cell cycle progression. HeLa, HEK293 and NB4 cells were each treated for 24 hr, either with DMSO (control), or with hinokiflavone, at a final concentration of 10 µM, 20 µM, or 30 µM. In the case of NB4 cells, the lower hinokiflavone concentrations of 0.5 µM, 1 µM, 2.5 µM and 5 µM were also tested. The cells were then fixed, labelled with propidium iodide and analyzed by flow cytometry (*Figure 4*). Interestingly, hinokiflavone differentially affected the cell lines tested, with most showing either cell cycle arrest, and/or eventual cell death, dependent upon concentration. The most dramatic effect, however, was observed for the acute promyelocytic cell line NB4, where most cells became apoptotic after 24 hr exposure to 10 µM hinokiflavone.

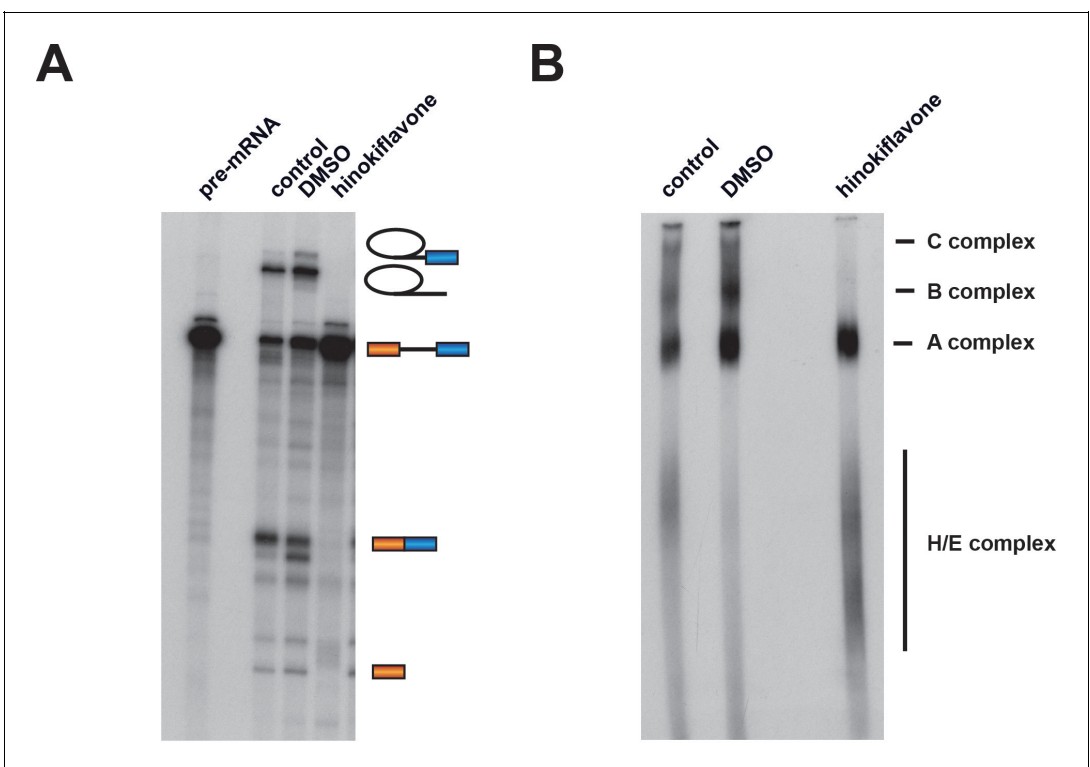

**Figure 3.** Hinokiflavone blocks spliceosome assembly prior to B complex formation. Formation of splicing complexes on the Ad1 pre-mRNA was analysed on a native agarose gel after incubation with either DMSO (control), or 500 µM hinokiflavone. The positions of the splicing complexes C, B, A and H/E are indicated on the right.

DOI: https://doi.org/10.7554/eLife.27402.008

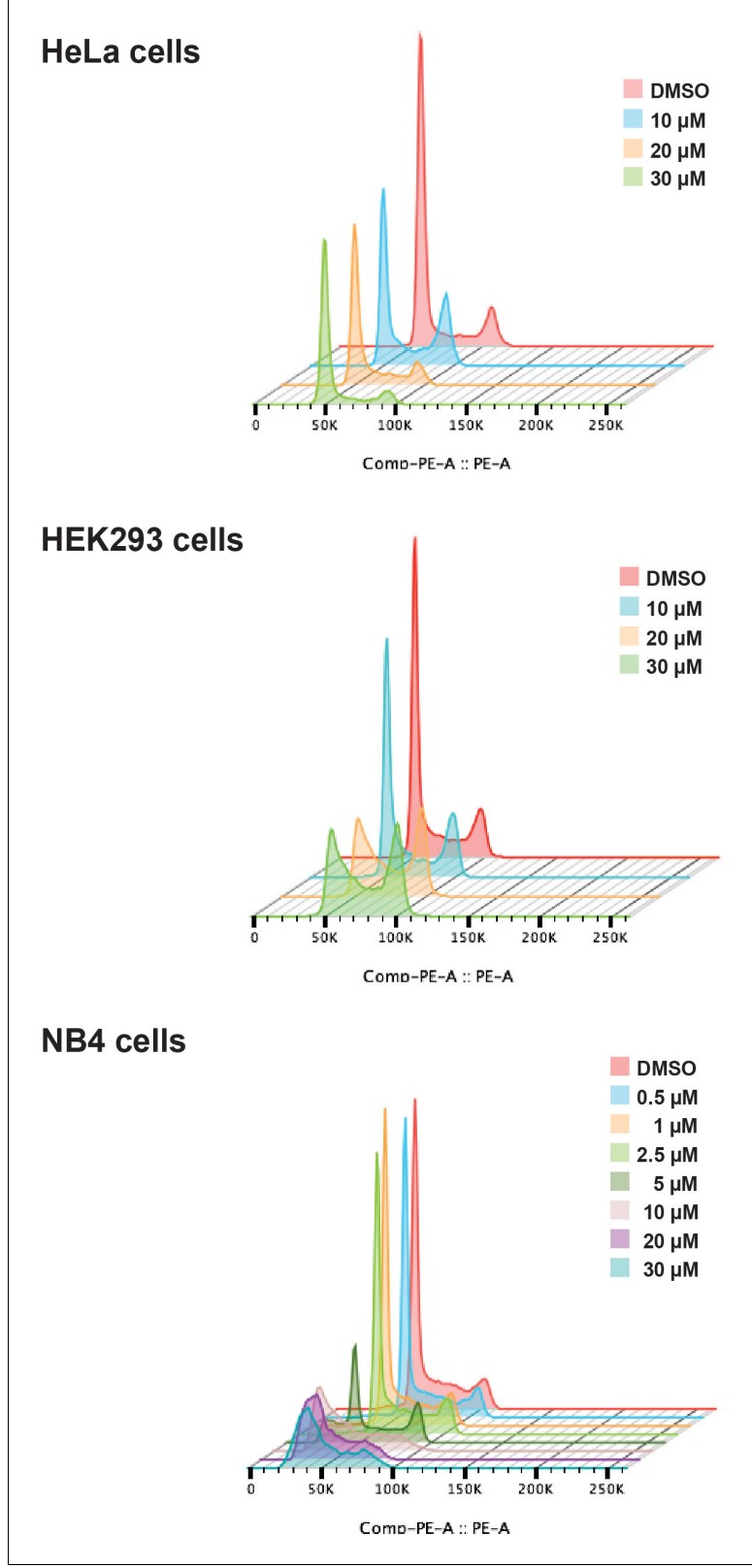

**Figure 4.** Hinokiflavone shows cell cycle specific effects. Cell cycle analysis was performed on HeLa, HEK293 and NB4 cells treated with either different concentrations of hinokiflavone, or DMSO (control), for 24 hr. Cellular DNA content was measured by propodium iodide staining followed by flow cytometry analysis.
DOI: https://doi.org/10.7554/eLife.27402.009

## Hinokiflavone alters nuclear organization of a subset of splicing factors

We examined the effect of hinokiflavone treatment on subcellular organization, in particular, the subnuclear organization of splicing factors and other nuclear components. For this, HeLa cells were treated with 20 µM hinokiflavone for 24 hr, fixed, permeabilised and stained with antibodies specific for the splicing factors SRSF2 (SC35), U1A, DDX46, U2AF65, SF3B1, SR proteins, CDC5L, PLRG1, BCAS2, PRP19, CTNNBL1 and snRNP200 (*Figure 5*). This showed a change in the 'speckled' nuclear staining pattern typical of many splicing factors, with the formation of enlarged and rounded 'mega speckles' (*Figure 5A*). Variation in the size (~0.5–4 µm) and number (~10–30) of mega speckles was observed between cells and the extent of mega speckle formation was dependent upon hinokiflavone concentration and length of treatment (*Figure 5—figure supplement 1*).

Interestingly, two groups of splicing factors could be distinguished, based upon their response to hinokiflavone. The first group, including SRSF2, U1A, DDX46, U2AF65, SF3B1 and SR proteins, which are all involved in early steps of spliceosome assembly, showed near complete relocalization into the mega speckles. However, a second group, including CDC5L, PLRG1, BCAS2, PRP19, CTNNBL1 and snRNP200, which are all associated with later steps of the splicing process and assemble into the spliceosome after A complex formation, were not enriched in the mega speckles, but instead retained a widespread nucleoplasmic distribution, similar to their localisation in control cells (*Figure 5B*). We note this differential localization response to hinokiflavone treatment for the two sets of splicing proteins closely matches the observed inhibition of spliceosome formation at the A complex observed with *in vitro* splicing extracts (*Figure 3B*). In contrast with these changes in nuclear organization, we observed little or no effect of hinokiflavone treatment on either cytoplasmic structures, or on the localization of multiple cytoplasmic proteins (data not shown).

HeLa cells treated with 20 µM hinokiflavone for 24 hr also showed a loss of Cajal bodies (CBs) in the nucleus. CBs play an important part in the maturation and recycling of snRNPs and typically accumulate splicing snRNPs and nucleolar snoRNPs, but not other protein splicing factors. Seven CB components were investigated, that is, coilin, SMN, TMG-cap, Y12, SNRPA1, CDK and Fibrillarin. Consistent with previous observations, in the control, DMSO-treated HeLa cells, coilin was enriched specifically in several CB foci, while SMN was observed in CBs in the nucleus and in punctate cytoplasmic structures. Components of splicing snRNPs, including TMG-cap, Y12 and SNRPA1, were located in both nuclear splicing speckles and CBs, while CDK and Fibrillarin localized in CBs and in the nucleolus (*Figure 6*). After treatment with hinokiflavone, bright CB foci were no longer observed, with coilin, SMN, TMG-cap, Y12 and SNRPA1 all relocalized and enriched in mega speckles. However, the cytoplasmic pool of SMN remained and appeared unaffected by hinokiflavone. Neither CDK, nor Fibrillarin, relocalized to the mega speckles, but the nucleoplasmic component of the staining pattern for both proteins was lost after treatment and only nucleolar CDK and Fibrillarin could be detected.

Disruption of CBs and relocalization of snRNP proteins is known to result from inhibition of transcription. We therefore compared the effect of hinokiflavone treatment with drugs that inhibit transcription, such as 5,6 Dichloro-1-beta ribofuranosylbenzimidazole (DRB), which inhibits transcription elongation by RNA polymerase II, using a fluorescence microscopy assay to detect transcription sites. In contrast with the major inhibition of pre-mRNA synthesis caused by DRB (*Figure 7A*), we observed little or no change in RNA synthesis levels in HeLa cells, either 4 hr or 8 hr after treatment with either 10 µM, 20 µM, or 30 µM hinokiflavone (*Figure 7A*). However, we observed a dose-dependent nuclear relocalization of SNRPA1, coilin, TMG-cap and SMN into mega speckles and disruption of Cajal bodies (*Figure 7B*). While TMG-cap and SNRPA1 relocated to mega speckles after treatment with DRB, coilin and SMN did not (*Figure 7B*). Coilin relocalized to the periphery of nucleoli and SMN dispersed as dots throughout the nucleoplasm. We conclude that the changes in subnuclear organisation induced by hinokiflavone are not likely to be only indirect effects of inhibiting transcription.

## Mega speckles contain polyadenylated RNA

Previous studies showed that treatment of cells with either SF3B1 inhibitors, such as meayamycin, spliceostatin, or platinolide B, or the inhibition of U4 snRNA, each lead to the accumulation of polyadenylated RNA in enlarged splicing speckles (*Hett and West, 2014*; *Kaida et al., 2007*). We therefore investigated whether the mega speckles induced by hinokiflavone also contained

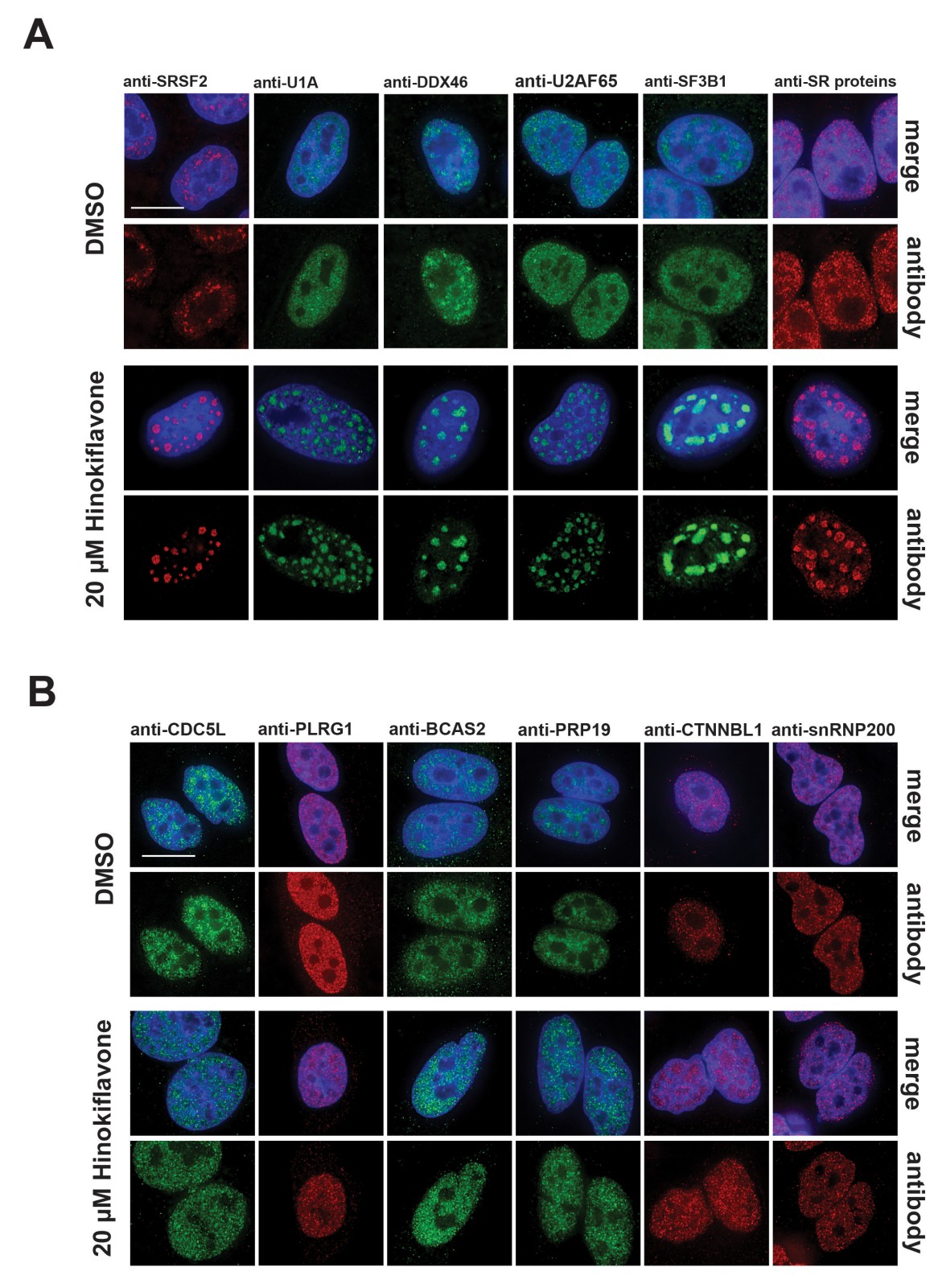

**Figure 5.** Changes in splicing speckles after treatment with hinokiflavone. HeLa cells incubated for 24 hr with either DMSO (control), or 20 µM hinokiflavone, were fixed and stained with antibodies for the following splicing factors: SRSF2, U1A, U2AF65, DDX46, SF3B1, SR proteins, CDC5L, PLRG1, BCAS2, PRP19, snRNP200 and CTNNBL1. (**A**) Splicing factors involved in the early steps of spliceosome assembly are located in enlarged

*Figure 5 continued on next page*

*Figure 5 continued*

splicing speckles. (B) Splicing factors that assemble after A complex formation are not accumulated in enlarged speckles and show a more diffuse nucleoplasmic distribution. Scale bars, 15 μm.

DOI: https://doi.org/10.7554/eLife.27402.010

The following figure supplement is available for figure 5:

**Figure supplement 1.** Mega speckles vary in size and number.

DOI: https://doi.org/10.7554/eLife.27402.011

polyadenylated RNA. To test this, we treated HeLa cells for 4 hr, 8 hr, or 24 hr with either different concentrations of hinokiflavone, or with DMSO (control) and investigated the localization of polyadenylated RNA by fluorescence in situ hybridization (*Figure 8*). This shows a hinokiflavone dose- and time-dependent loss of the cytoplasmic polyadenylated RNA signal (*Figure 8*, arrowheads indicate cytoplasmic signal) and a concomitant accumulation of polyadenylated RNA in the nuclear mega speckles. Interestingly, we observe that after 24 hr treatment with 30 μM hinokiflavone, the polyadenylated RNA is concentrated in small spots at the periphery of the mega speckles (*Figure 8—figure supplement 1*).

## Hinokiflavone promotes nuclear relocalization of SUMO

Due to the unexpected changes in the localization of coilin and SMN induced by hinokiflavone, we next tested whether the localization of other nuclear proteins, including factors that are not

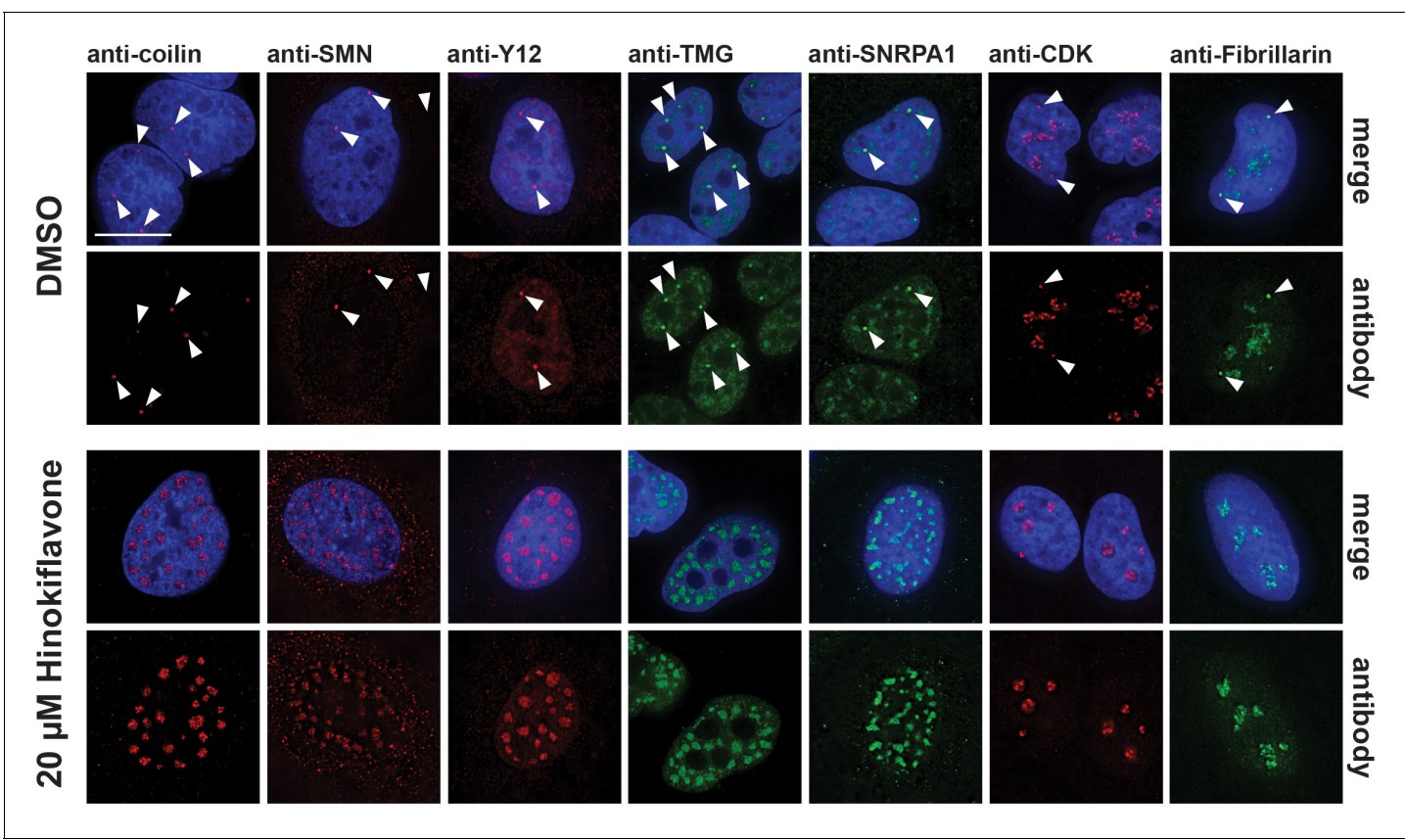

**Figure 6.** Hinokiflavone treatment leads to relocation of CB components to mega speckles. HeLa cells were treated for 24 hr with either DMSO (control), or 20 μM hinokiflavone and the fixed cells were stained with anti-coilin, anti-SMN, anti-Y12, anti-TMG, anti-SNRPA1, anti-CDK and anti-Fibrillarin antibodies, respectively. Coilin, SMN, Y12 and TMG show relocation to the enlarged speckles containing splicing factors in hinokiflavone treated cells. Arrowheads denote intact CBs. CDK and Fibrillarin are only detected in nucleoli after treatment with hinokiflavone. Scale bars, 15 μm.

DOI: https://doi.org/10.7554/eLife.27402.012

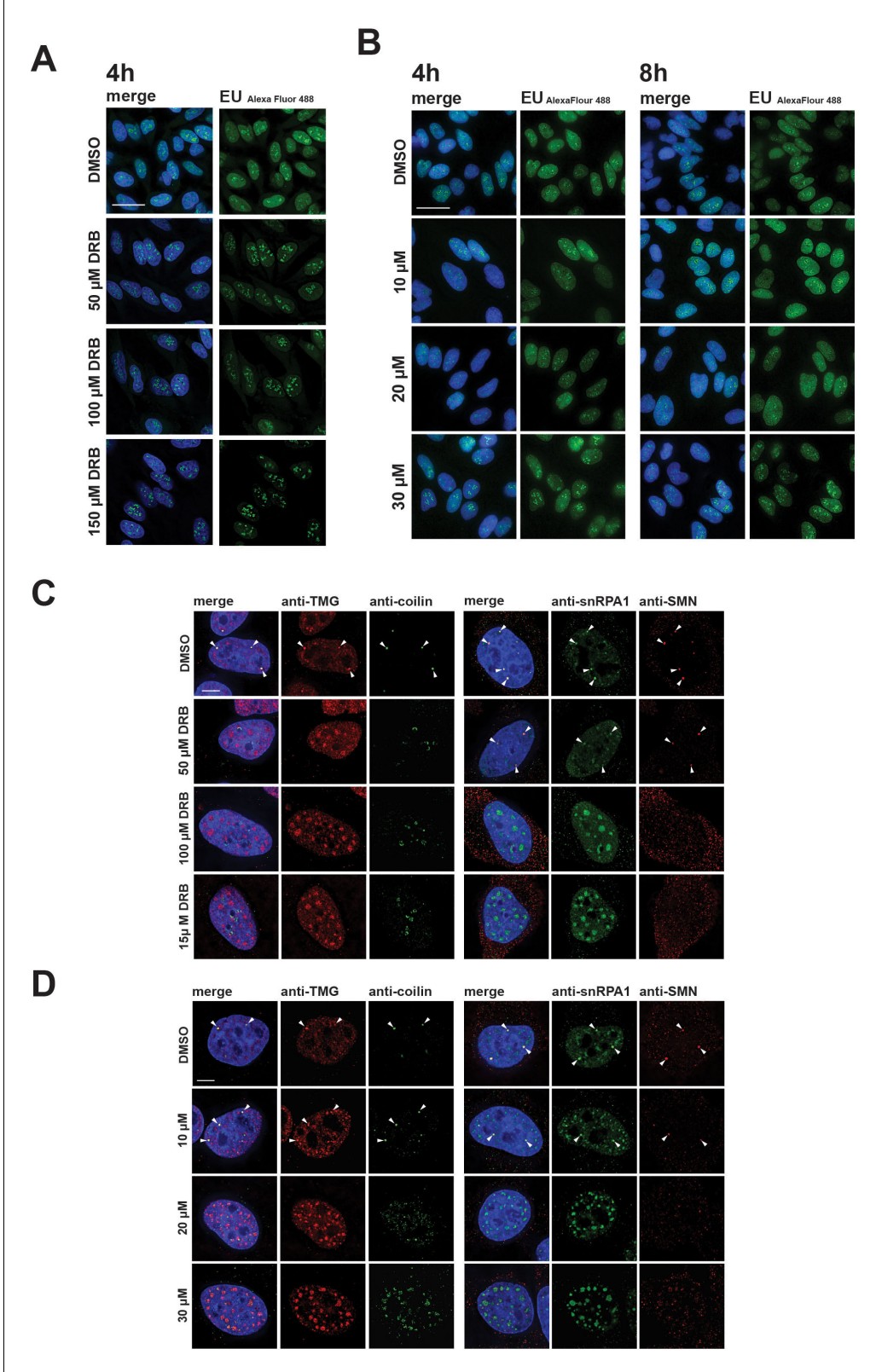

**Figure 7.** Differential effects of hinokiflavone and DRB on inhibition of RNA polymerase II transcription and disruption of Cajal bodies. HeLa cells were incubated for either 4 hr with DRB (**A**), or 4 hr and 8 hr with hinokiflavone (**B**), before cells were incubated with EU to label newly synthesized RNA. Cells were fixed and labelled RNA was detected by fluorescence microscopy. Scale bar, 40 μm. (**C**) HeLa cells were incubated with either DMSO (control), or with 50 μM, 100 μM, or 150 μM DRB, for 4 hr, then stained with anti-coilin, anti-TMG, anti-SMN and anti-SNRPA1 antibodies, respectively. Arrowheads

*Figure 7 continued on next page*

*Figure 7 continued*

denote intact CBs. (**D**) HeLa cells were incubated with either DMSO (control), or with 10 μM, 20 μM, or 30 μM hinokiflavone, for 8 hr, then fixed and stained with anti-coilin, anti-TMG, anti-SMN and anti-SNRPA1 antibodies, respectively. Arrowheads denote intact CBs. Scale bar, 6.5 μM.
DOI: https://doi.org/10.7554/eLife.27402.013

spliceosome components, such as PML, the cleavage stimulation factor subunit 2 (CSTF2), SUMO1 and SUMO2/3, was affected. After treatment with hinokiflavone, we observed a loss of PML-containing nuclear bodies (PML-NBs), with PML relocalized into a pattern of dots, situated at the periphery of the enlarged mega speckles containing splicing factors (*Figure 9A*). The diffuse nucleoplasmic localization of CSTF2 seen in control cells was lost, with CSTF2 also relocated to dots at the periphery of the mega speckles (*Figure 9B*).

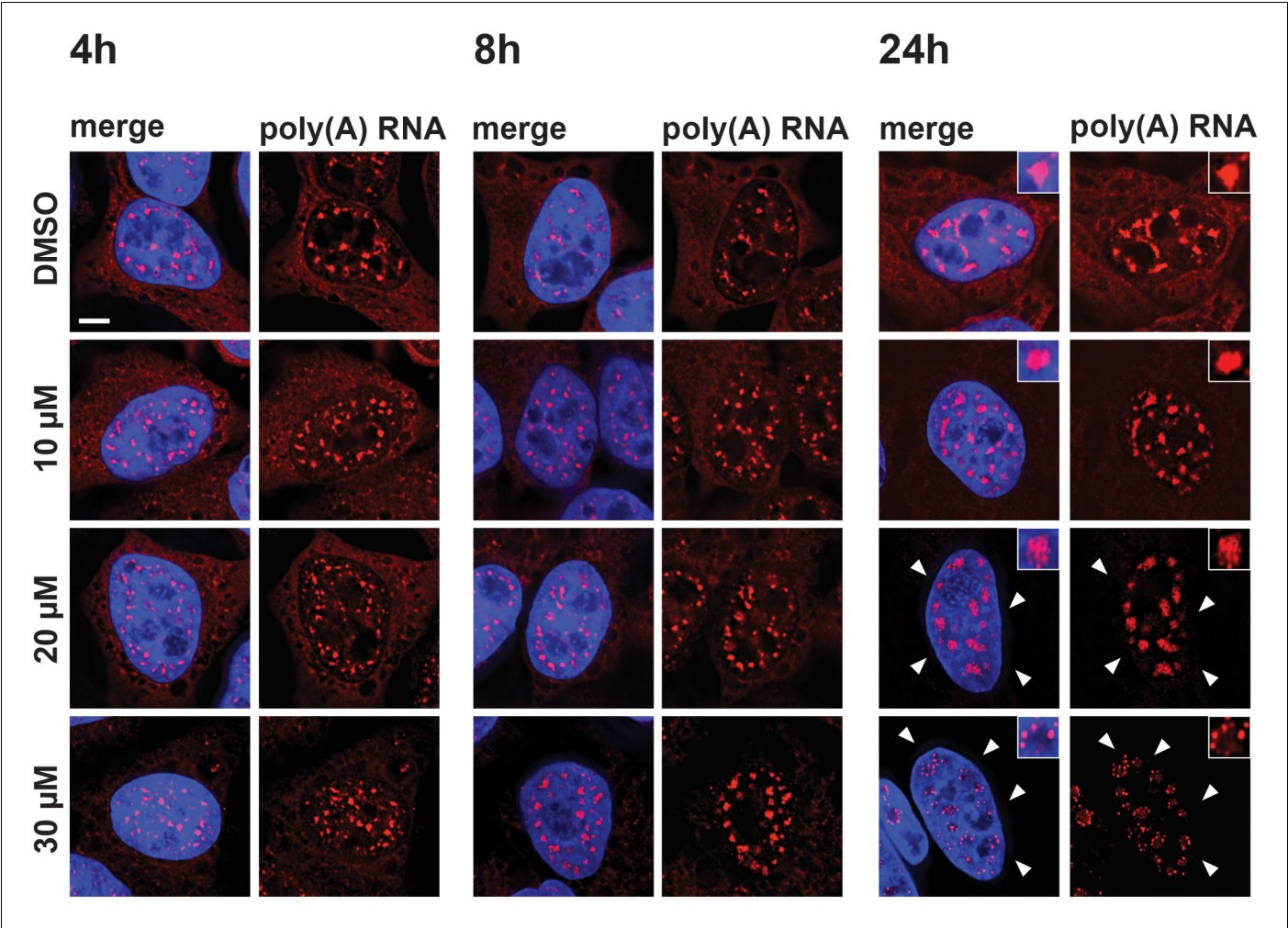

**Figure 8.** Hinokiflavone affects localization of polyadenylated RNA. HeLa cells were treated for 4 hr, 8 hr, or 24 hr, with either DMSO (control), or 10 μM, 20 μM, or 30 μM hinokiflavone and the fixed cells were hybridized with Cy3 labelled Oligo dT probes. After 24 hr the poly(A) RNA is lost from the cytoplasm (arrowheads) in the presence of 20 and 30 μM hinokiflavone. Treatment with 30 μM hinokiflavone leads to a relocation of poly(A) RNA from the enlarged splicing speckles to circles of dots. Scale bars, 6.5 μm.
DOI: https://doi.org/10.7554/eLife.27402.014
The following figure supplement is available for figure 8:

**Figure supplement 1.** Hinokiflavone affects localization of polyadenylated RNA.
DOI: https://doi.org/10.7554/eLife.27402.015

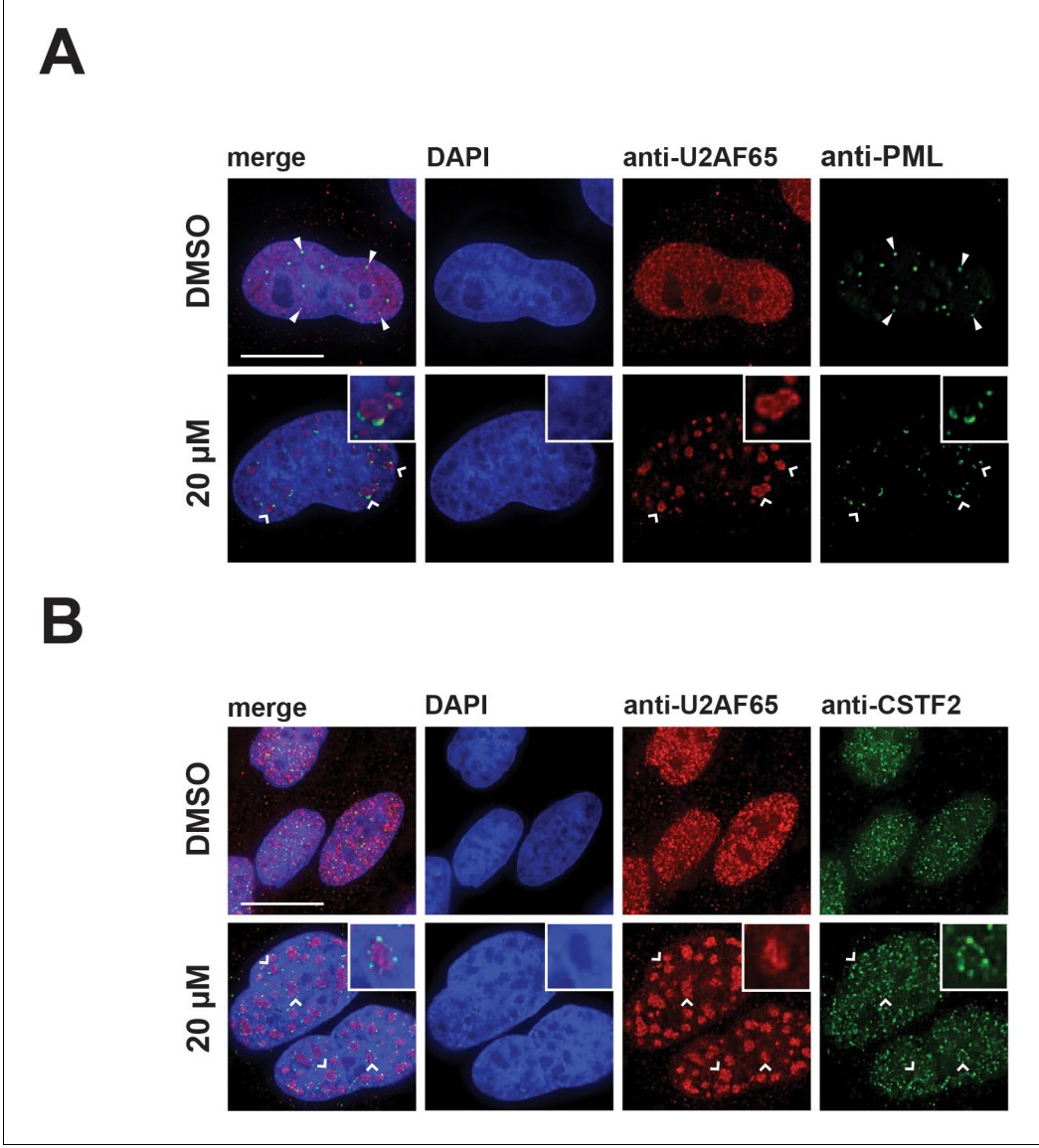

**Figure 9.** Hinokiflavone treatment leads to the relocation of nuclear proteins to the periphery of enlarged splicing speckles. HeLa cells were treated for 24 hr with either DMSO (control), or 20 μM hinokiflavone and the fixed cells were stained with either (**A**) anti-U2AF65 and anti-PML, or (**B**) anti-CSTF2 antibodies. Co-staining with anti-U2AF65 antibodies showed that both CSTF2 and PML relocate to the periphery of enlarged splicing speckles (highlighted by carets and enlarged images). Arrowheads denote PML bodies. Scale bars, 15 μm.
DOI: https://doi.org/10.7554/eLife.27402.016

In the DMSO treated control cells, SUMO2/3 showed diffuse nucleoplasmic staining and bright foci, while SUMO1 showed both staining at the nuclear membrane and concentration in nucleoplasmic foci. Surprisingly, we observed that both SUMO1 and SUMO2/3 also relocalized to mega speckles in hinokiflavone-treated cells (*Figure 10A*). This was also seen in HeLa cells treated for 2 hr with 10 μM, 20 μM, or 30 μM hinokiflavone, and stained with antibodies specific for either SUMO1, SUMO2/3, or SRSF2 (SC35). Already after this short, 2 hr exposure to hinokiflavone, all three markers showed a large-scale relocalization into mega speckles (*Figure 10B*).

## Hinokiflavone increases levels of SUMOylated proteins

Following the observed relocalization of SUMO1 and SUMO2/3 to mega speckles, we next tested whether hinokiflavone treatment altered the protein SUMOylation pattern in cells. For this, HEK293

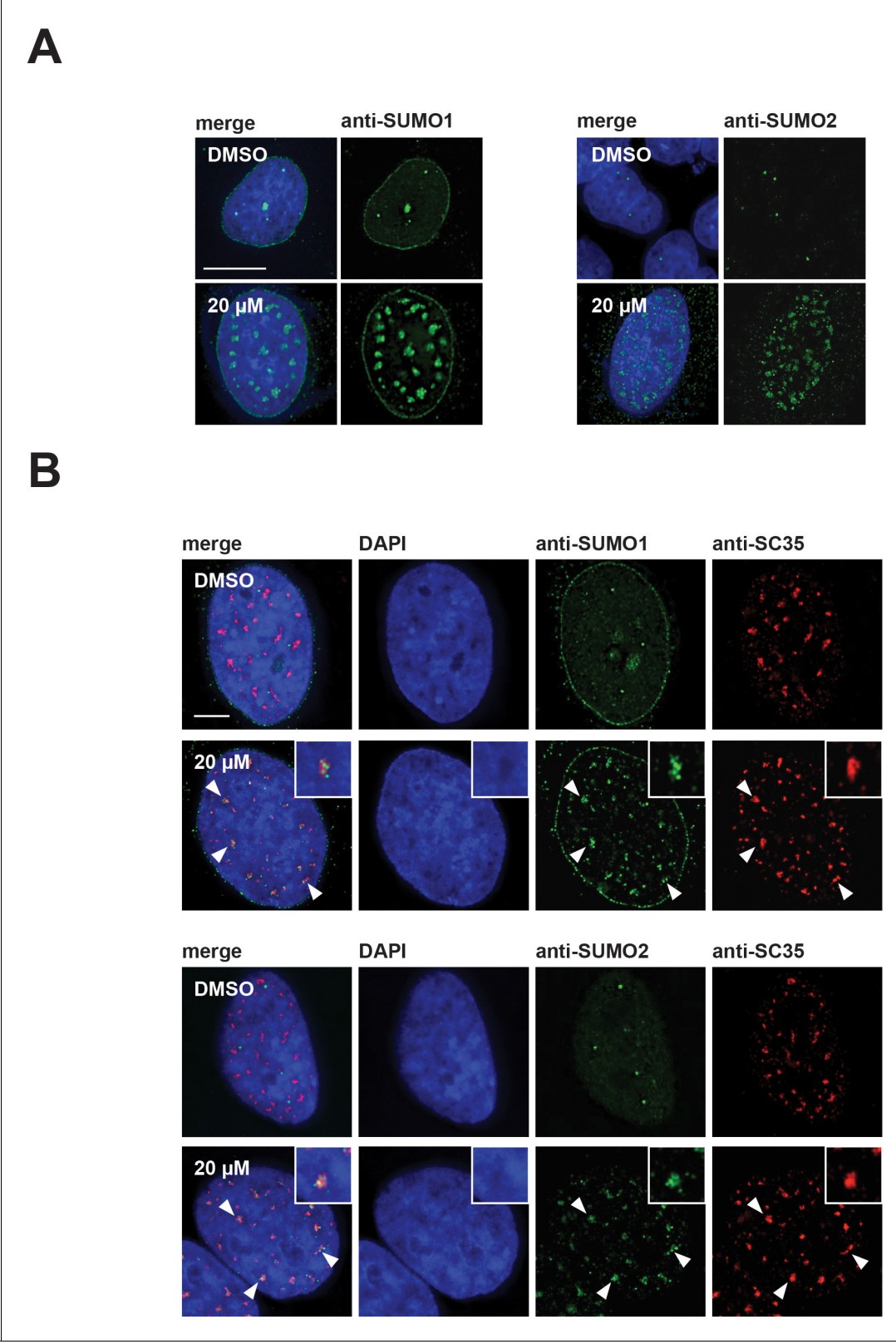

**Figure 10.** SUMO1 and SUMO2/3 relocalize to enlarged splicing speckles in the presence of hinokiflavone. (**A**) HeLa cells were treated for 24 hr with either DMSO (control), or 20 µM hinokiflavone and the fixed cells were stained with either anti-SUMO1, or anti-SUMO2/3 antibodies. Both SUMO1 and SUMO2/3 accumulated in the enlarged splicing speckles formed after treatment with hinokiflavone. (**B**) Treatment of HeLa cells with 20 µM

*Figure 10 continued on next page*

*Figure 10 continued*
hinokiflavone for 2 hr. Co-staining with either anti-SUMO1, or anti-SUMO2/3 and anti-SRSF2 (SC35) antibodies showed that both SUMO1 and SUMO2/3 accumulate in enlarged splicing speckles (highlighted by arrowheads and enlarged images). Scale bars, 15 μm.
DOI: https://doi.org/10.7554/eLife.27402.017

cells were treated for 24 hr, either with DMSO (control), or with 10 μM, 20 μM, or 30 μM hinokiflavone, followed by harvesting, preparation of total cell lysates, separation of proteins by SDS-PAGE and analysis by immunoblotting (*Figure 11*). This shows a clear increase in the accumulation of high molecular weight, SUMO1 and SUMO2/3-modified proteins in the hinokiflavone treated extracts, as compared with the DMSO control. Interestingly, parallel immunoblotting analysis for the related peptide modifiers ubiquitin and NEDD8, showed no increase in their levels of protein conjugation in extracts from hinokiflavone treated cells, but rather a modest decrease. This demonstrates a remarkably selective effect of hinokiflavone for promoting specifically increased levels of protein SUMOylation *in cellulo*. We also observed an increase in protein SUMO2/3 SUMOylation after hinokiflavone treatment of HeLa and NB4 cells (*Figure 11—figure supplement 1*).

We next analyzed the effect of hinokiflavone on protein SUMOylation levels in the nuclear extracts used for in vitro splicing reactions. First, to evaluate the concentrations of hinokiflavone that inhibit pre-mRNA splicing in vitro, HeLa cell nuclear extract was treated either with DMSO (control), or with concentrations of hinokiflavone between 25 μM - 500 μM. The lowest concentration at which splicing inhibition was detected was 50 μM (*Figure 12A*).

To examine the effect of hinokiflavone on protein SUMOylation levels in in vitro splicing reactions, a final concentration of either 100 μM, 300 μM, or 500 μM hinokiflavone was used. After incubation at 30°C for 90 min, the splicing reactions were separated by SDS-PAGE and immuno-blotted to detect either SUMO1, SUMO2/3, or SRSF1, the latter acting as a loading control (*Figure 12B*). In the presence of hinokiflavone we observed a major increase in the levels of high molecular weight, SUMO1 and SUMO2/3-modified proteins, when compared with the DMSO control (*Figure 12B*). Furthermore, this increase in protein SUMOylation levels is hinokiflavone concentration-dependent.

In summary, we conclude that hinokiflavone causes an increase in protein SUMOylation levels, both *in cellulo* and in vitro.

## Hinokiflavone inhibits SENP1 activity

Protein SUMOylation is a reversible modification. The level of SUMOylated protein thus reflects the balance between the rate of SUMO conjugation by the SUMO conjugation machinery and the rate of SUMO deconjugation, catalyzed by sentrin-specific proteases (SENPs). We therefore hypothesized that the observed increase in protein SUMOylation caused by hinokiflavone, both *in cellulo* and in vitro, could result from an inhibition of SENP activity. To test this hypothesis, we performed in vitro SENP assays, using a purified, catalytically active fragment of the SENP1 protein (aa 415–643), which was expressed in *E.coli* (*Figure 13A*). We compared SENP1 activity when incubated in the presence of DMSO (control), or 500 μM of either hinokiflavone, or the four other biflavones previously tested for their ability to inhibit splicing (i.e., amentoflavone, cupressuflavone, isoginkgetin and sciadopitysin). The SENP1 assays were carried out as described in experimental procedures, proteins were separated using a 4–12% Bis-Tris PAGE gel and proteins were visualized by staining with Coomassie blue (*Figure 13A*). This showed a clear inhibition of SENP1 activity by hinokiflavone (lane 5), as compared with the DMSO control (lane 4). Interestingly, SENP1 inhibition was also seen with some of the other biflavones, with the degree of SENP1 inhibition correlated with their ability to inhibit pre-mRNA splicing in vitro (cf. *Figure 1B*). Thus, hinokiflavone and amentoflavone show the greatest inhibitory effect in vitro on both splicing and SENP1 activity.

The drug affinity responsive target stability (DARTS) assay provides a convenient way of testing whether a compound binds to putative target proteins (*Pai et al., 2015*; *Lomenick et al., 2011*). We therefore used the DARTS approach to evaluate whether hinokiflavone can bind to recombinant SENP1. For this assay, the purified, catalytically active SENP1 protein fragment was incubated with either DMSO (control), or 500 μM hinokiflavone, before the samples were subjected to limited digestion with varying concentrations of pronase, from ~16–200 μg/ml (*Figure 13B*). This showed that in

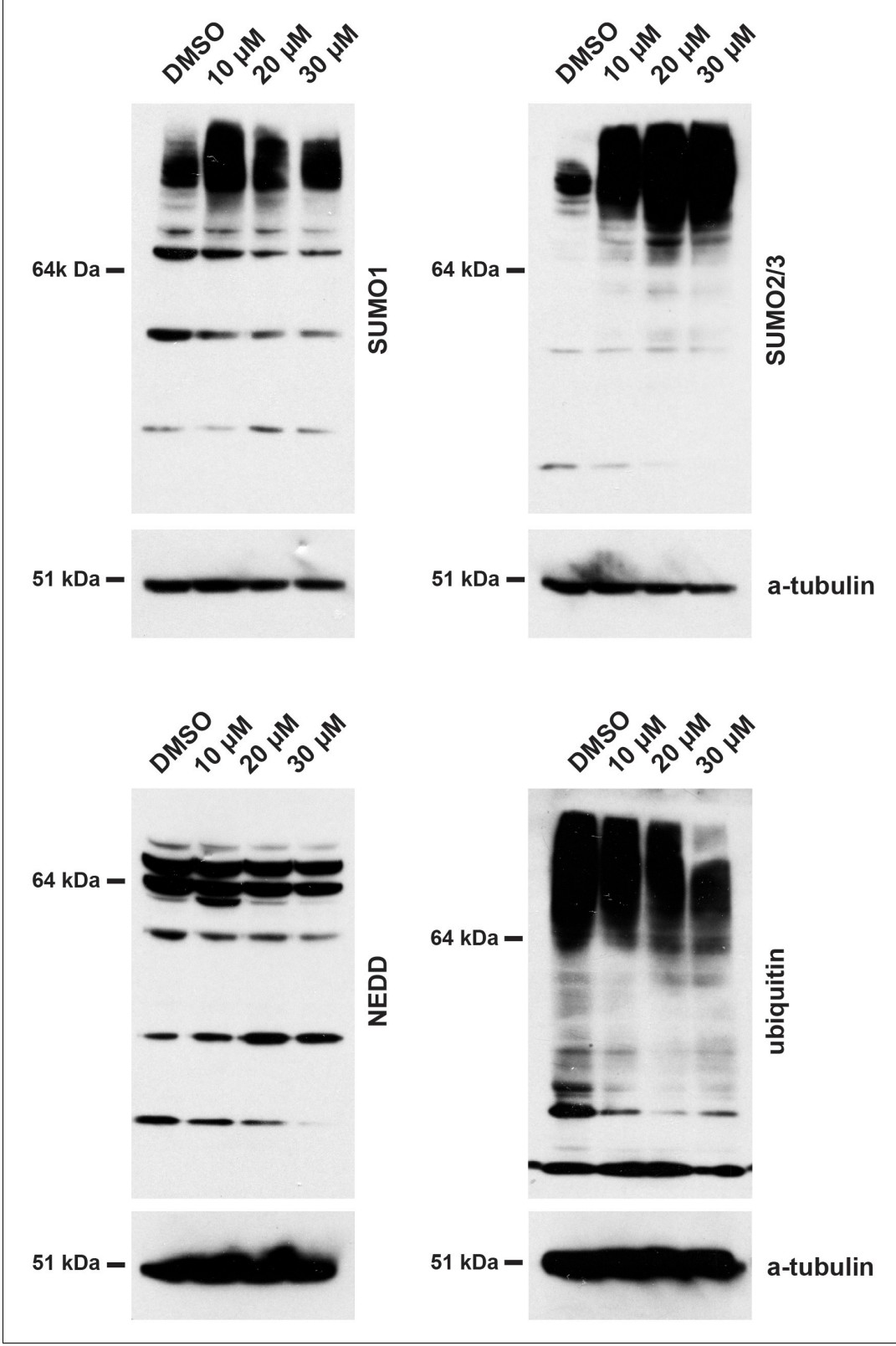

**Figure 11.** Treatment of HEK293 cells with hinokiflavone leads to an increase in SUMOylated proteins. HEK293 cells were treated with either DMSO (control), or with 10 µM, 20 µM, or 30 µM hinokiflavone for 24 hr before cells were lysed in 1x LDS buffer. Samples were separated on SDS-PAGE and transferred to membranes. After probing with antibodies specific for SUMO1, SUMO2/3, Ubiquitin, and NEDD8, labelled proteins were visualized using

*Figure 11 continued on next page*

*Figure 11 continued*

chemiluminescence, showing a specific accumulation of poly-SUMOylated proteins after hinokiflavone treatment. The membranes were also probed to detect alpha-tubulin as a control (bottom panels).

DOI: https://doi.org/10.7554/eLife.27402.018

The following figure supplement is available for figure 11:

**Figure supplement 1.** Hinokiflavone treatment leads to an increase in SUMO2/3 modified proteins in HeLa and NB4 cells.

DOI: https://doi.org/10.7554/eLife.27402.019

the presence of hinokiflavone, the sensitivity of the SENP1 fragment to protease digestion increased, indicating that hinokiflavone directly interacts with SENP1.

Another method to analyze ligand target interactions is the cellular thermal shift assay (CESTA). HeLa nuclear extract was incubated with either DMSO (control), or 500 µM hinokiflavone and heated to different temperatures, ranging from 30–50°C. The presence of soluble, endogenous SENP1 at each temperature was examined by western blotting (*Figure 13C*). In the presence of DMSO SENP1 was still soluble at 43°C, whereas no soluble SENP1 could be detected at any of the temperatures tested in the presence of hinokiflavone. This also indicates that hinokiflavone binds directly to SENP1, thereby altering its structure and concomitantly decreasing its thermal stability, consistent with the increase in protease sensitivity seen in the DARTS assay.

In summary, these data show that a structurally related group of biflavone compounds can inhibit purified SENP1 SUMO protease in vitro and this correlates with their potency as in vitro inhibitors of pre-mRNA splicing. The strongest inhibitor, hinokiflavone, was further shown to interact with SENP1 in vitro, using both the DARTS and CESTA assays. These data support the hypothesis that cells treated with hinokiflavone are prevented from de-conjugating SUMO, resulting in an accumulation of polySUMOylated proteins to levels above that seen in control cells.

## Identification of protein SUMOylation targets affected by hinokiflavone

Having identified hinokiflavone as a SUMO protease inhibitor in vitro, which causes the accumulation of poly-SUMOylated proteins *in cellulo*, we next sought to identify protein targets whose level of SUMO modification increases in cells treated with hinokiflavone. For this analysis, we used a previously described, quantitative, SILAC proteomics approach in HEK293 SUMO2$^{T90K}$ cells (*Tammsalu et al., 2014*). Thus, HEK293 SUMO2$^{T90K}$ cells, which express the His-tagged SUMO2$^{T90K}$ mutant, were treated for 8 hr with either DMSO (control), or with 20 µM hinokiflavone. After harvesting, an equal amount of each cell pellet was combined and a lysate was prepared. SUMO2$^{T90K}$-conjugated target proteins in the lysates were affinity purified under denaturing conditions using the His-tag. After cleavage with endonuclease Lys-C, an antibody specific for the diGly-modified Lys peptide, which is diagnostic of SUMO2$^{T90K}$ modification, was used to enrich for peptides including lysine residues that had been conjugated to SUMO2$^{T90K}$.

This analysis identified 924 SUMO2-modified lysine residues in 543 different proteins (*Supplementary file 1*; Raw MS data available from PRIDE repository, accession number PXD007629). Of these, twenty-two lysine residues showed more than a five-fold increase in the level of SUMO2 modification in the hinokiflavone-treated cells, when compared with the DMSO control. Interestingly, ten of these sites are located in six proteins (*Table 1*) that are components of the U2 snRNP, that is, PRPF40A, SF3B2, SF3A2, SNRPD2 (SMD2), U2SURP (SR140) and SF3B1 (*Figure 14A*). These six proteins have previously been shown to interact, as illustrated in *Figure 14B*. The most dramatic effect of hinokiflavone was seen for the U2 snRNP associated protein PRPF40A, which increased SUMO2 modification levels at four lysine residues, that is, K241, K375, K517 and K707, including very large (>20 fold) increases in the level of SUMO2 at K241, K375 and K517, in the hinokiflavone-treated cells.

To validate these MS data, we performed immuno-blotting experiments, using an antibody specific for PRPF40A, with extracts prepared from HEK293 cells treated for 24 hr with either DMSO (control), or with 10 µM, 20 µM, or 30 µM hinokiflavone (*Figure 15B*). This resulted in higher molecular weight bands in the presence of hinokiflavone, migrating above the main band representing PRPF40A. To verify that these additional bands represent SUMO2 modified PRPF40A, HeLa cells

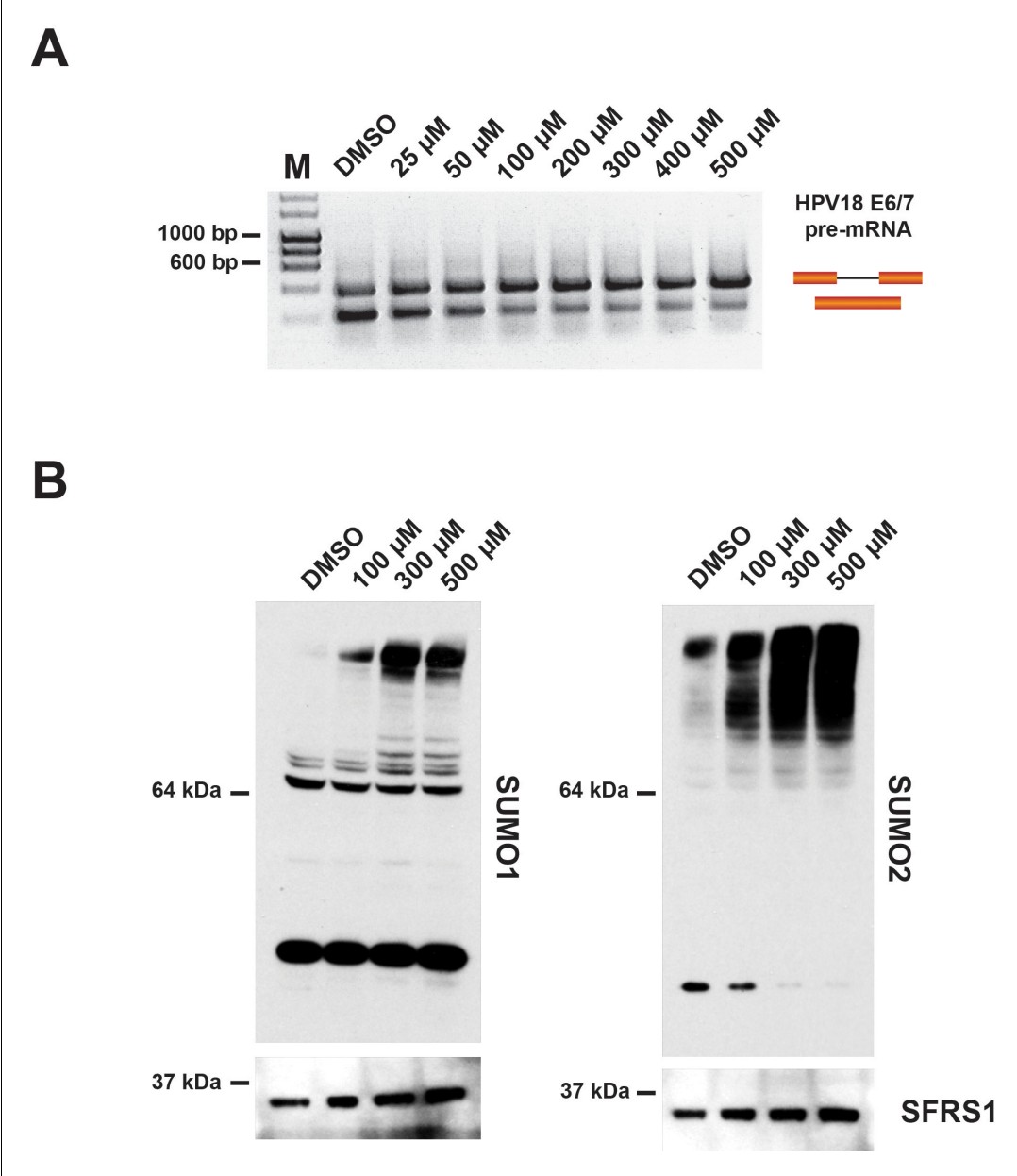

**Figure 12.** Incubation of in vitro nuclear splicing extracts with hinokiflavone leads to an increase in SUMOylated proteins. HeLa nuclear extract reactions were incubated in vitro under splicing conditions with either DMSO (control), or with increasing concentrations of hinokiflavone from 25 μM-500 μM. (**A**) Evaluation of the lowest concentration at which hinokiflavone inhibits pre-mRNA splicing in vitro. (**B**) Proteins were extracted and size-separated by SDS-PAGE, transferred to membranes and probed using either anti-SUMO1, anti-SUMO2/3, or anti-SRSF1 antibodies and visualized using chemiluminescence. A specific accumulation of hyper-SUMOylated proteins after hinokiflavone treatment is shown. Membranes were also probed to detect SFRS1 as a loading control (bottom panels).

DOI: https://doi.org/10.7554/eLife.27402.020

stably expressing YFP-SUMO2 were treated with either DMSO (control), or with 20 μM hinokiflavone, for 8 hr, before an immunoprecipitation of YFP-SUMO2 was performed, as described in the methods section. In comparison to the DMSO control, a dramatic increase in the level of SUMO2 modified PRPF40A was detected, representing >15% of the total PRPF40A (*Figure 15B*). In addition, immunofluorescence analysis, using the same antibody, showed that after hinokiflavone treatment PRPF40A accumulated in the mega speckles (*Figure 15A*).

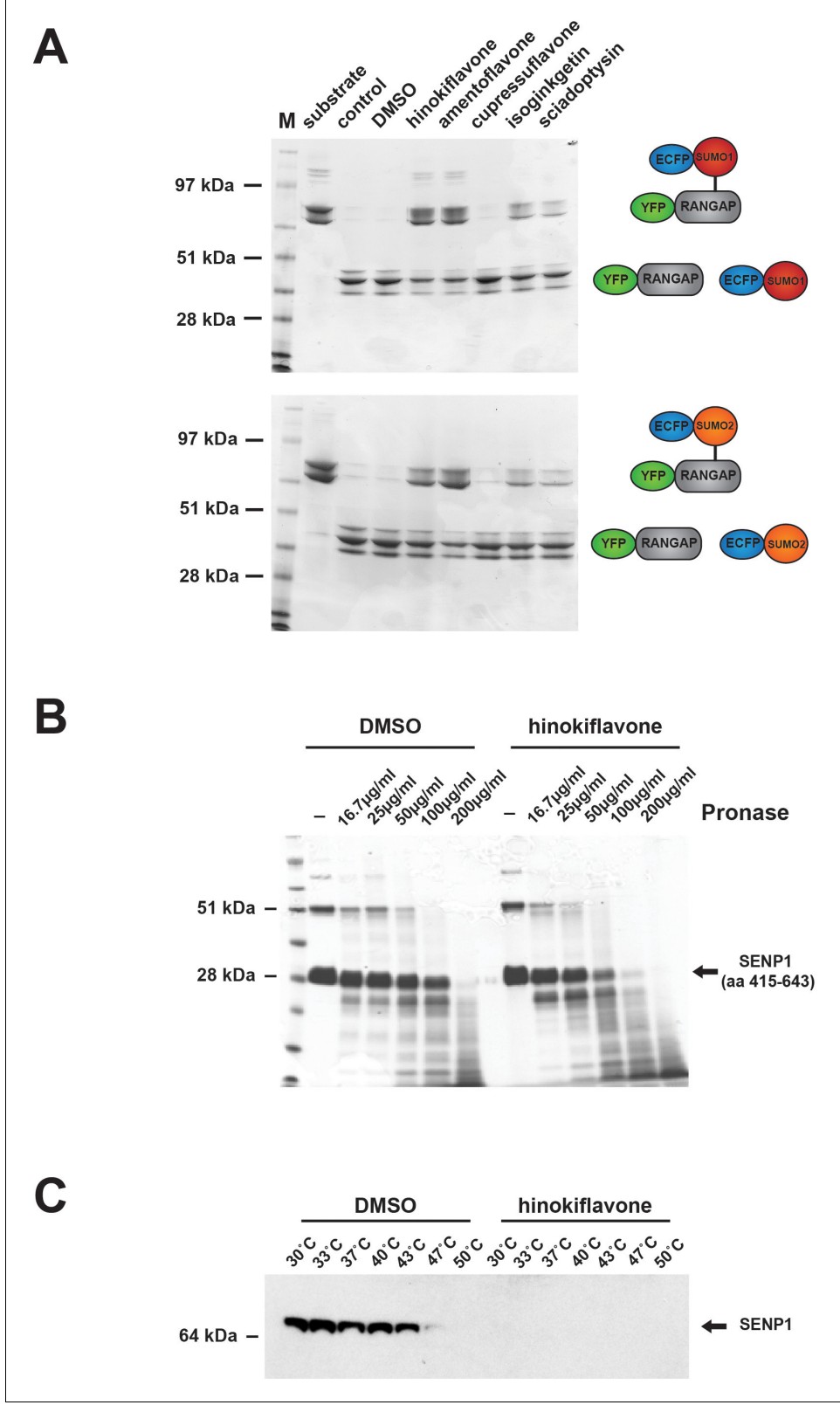

**Figure 13.** Biflavones inhibit SENP1 in vitro. (**A**) The effect of 500 μM hinokiflavone, amentoflavone, cupressuflavone, isoginkgetin and sciadopitysin on the isopeptidase activity of a highly purified fragment of catalytically active SENP1 (comprising aa 415–643), was determined by an in vitro gel-based activity assay. (**B**) DARTS Assay; incubation of the catalytically active SENP1 fragment with either DMSO (control), or 500 μM

*Figure 13 continued on next page*

*Figure 13 continued*

hinokiflavone, before the samples were digested with different concentrations (from ~16–200 µg/ml) of the proteinase pronase. SENP1 showed an increased sensitivity to protease digestion in the presence of hinokiflavone. (**C**) CESTA Assay; HeLa nuclear extract was treated with either DMSO, or 500 µM hinokiflavone for 20 min at RT, followed by heat treatment and ultracentrifugation. Western blot analysis of the soluble proteins demonstrated a dramatic change in the thermal stability of SENP1 in the presence of hinokiflavone.

DOI: https://doi.org/10.7554/eLife.27402.021

## SUMOylated splicing factors accumulate in the insoluble fraction in cell extracts treated with hinokiflavone

HEK293 cells stably expressing GFP-PRPF40A were established and the behaviour of GFP-PRPF40A in the presence of hinokiflavone was tested. Like endogenous PRPF40A, GFP-PRPF40A relocated to mega speckles after hinokiflavone treatment (*Figure 16A*). This cell line was then used to examine if the SUMOylation of PRPF40A changed its interactions with other splicing factors. GFP-PRPF40A expressing cells were thus treated with either DMSO (control), or 20 µM hinokiflavone, for 8 hr, before cells were lysed with Co-IP buffer. After centrifugation, the soluble GFP-tagged proteins were immunoprecipated using GFP-Trap beads and analyzed by western blotting (*Figure 16B*). Immunoprecipation of PRPF40A co-isolates SF3B2, but, as expected, not PRP19. However, SUMOylated PRPF40A and SF3B2 partitioned into the non-soluble fraction of hinokiflavone treated cells. Thus, with this approach it was not possible to determine whether SUMOylation affects protein-protein interactions for PRPF40A. In summary, the data show that hinokiflavone promotes hyper-SUMOylation of splicing factors, resulting in a major change in their biophysical properties, as reflected in the increased formation of insoluble aggregates in cell free extracts.

## Discussion

In this study, we have identified the plant biflavone, hinokiflavone, to be a novel modulator of pre-mRNA splicing activity, both in vitro and *in cellulo*. We confirmed this to be a specific effect of hinokiflavone by developing a route for making synthetic hinokiflavone, which was shown to be active. Hinokiflavone blocks splicing of pre-mRNA substrates in vitro by inhibiting spliceosome assembly, specifically preventing B complex formation.

Multiple human cell lines treated with hinokiflavone show changes in alternative splicing patterns and altered nuclear organization of splicing factors required for the early stages of spliceosome assembly leading to A complex formation. This results in disruption of Cajal bodies (CBs) and concomitant formation of 'mega speckles', i.e., enlarged nuclear splicing speckles enriched in a subset of spliceosome proteins, along with SUMO1, SUMO2 and polyadenylated RNA. Hinokiflavone treated cells show dose and time-dependent cell cycle arrest phenotypes, along with varying

**Table 1.** SUMO-modified lysine residues in U2 snRNP proteins upregulated by hinokiflavone.

| Gene name | Modified sequence | Position | Upregulation |
|---|---|---|---|
| PRPF40A | _SNLHAM(ox)IK(gl)AEESSK_ | 241 | 145.8 |
| | _DVLFFLSK(gl)K_ | 517 | 47.5 |
| | _TVADFTPK(gl)K_ | 375 | 45.5 |
| | _DFVAIISSTK(gl)RSTTLD_ | 707 | 7.8 |
| SF3B2 | _TGK(gl)PLYGDVFGTNAAE_ | 680 | 94.4 |
| | _M(ox)GK(gl)IDIDYQK_ | 563 | 56.5 |
| SF3A2 | _(ac)M(ox)DFQHRPGGK(gl)TGSGGVASSSE_ | 10 | 22.6 |
| SNRPD2 | _PK(gl)SEM(ox)TPEELQK_ | 8 | 16.2 |
| U2SURP | _HHLYSNPIK(gl)EE_ | 822 | 10.0 |
| SF3B1 | _GYK(gl)VLPPPAGYVPIRTPARK_ | 413 | 5.0 |

DOI: https://doi.org/10.7554/eLife.27402.023

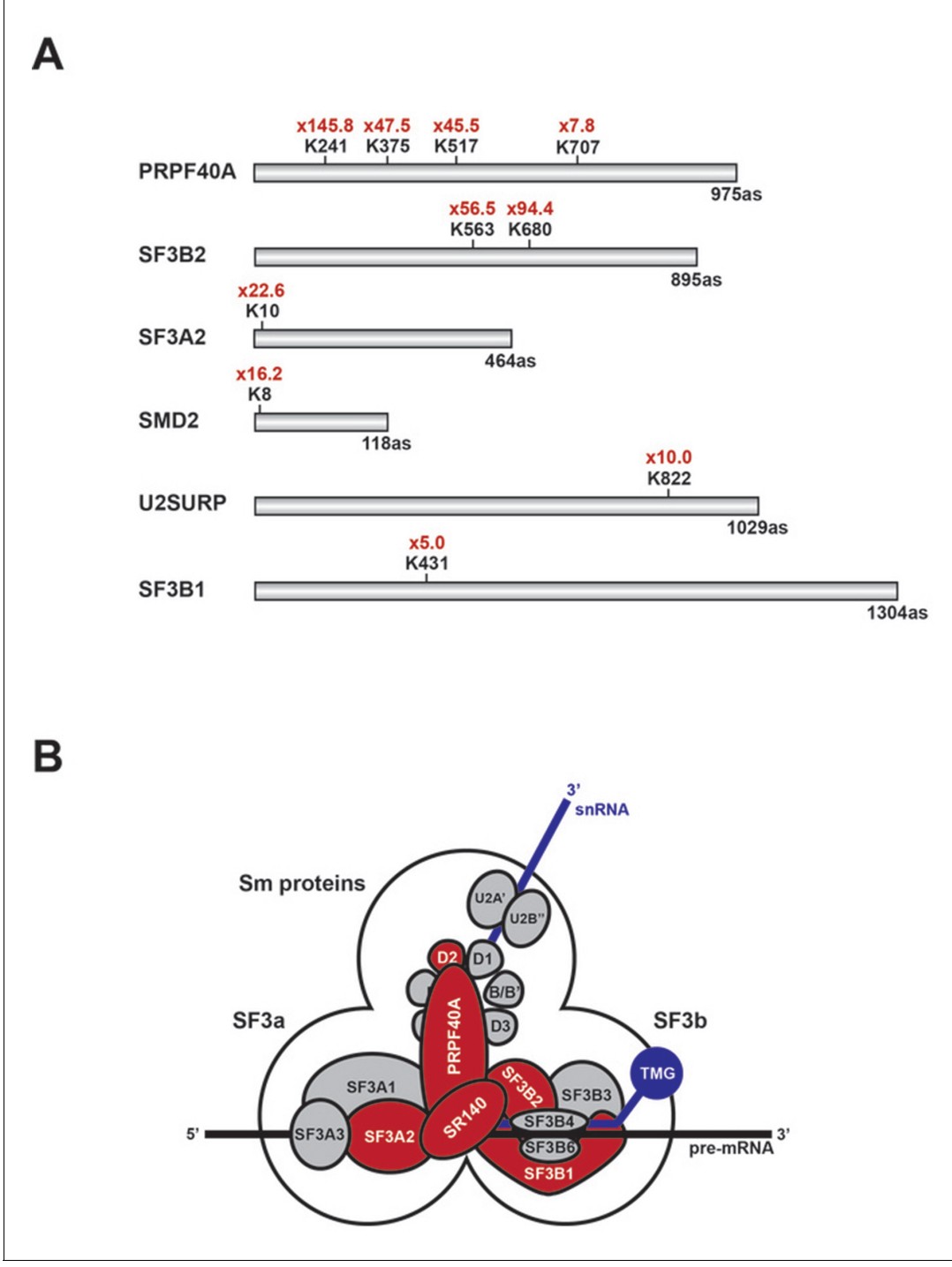

**Figure 14.** Schematic representation of splicing factors identified as SUMO2 target proteins. (**A**) Lysine residues in PRPF40A, SF3B2, SF3A2, SMD2, SR140 (U2SURP) and SF3B1 that show increased SUMO2 modifications in HEK293 cells treated for 8 hr with 20 µM hinokiflavone are shown. (**B**) Schematic representation of the SUMO2-modified U2 snRNP components, which are coloured in red.
DOI: https://doi.org/10.7554/eLife.27402.022

degrees of apoptosis, the latter effect seen most strikingly in the promyelocytic NB4 cell line that expresses the PML-RARalpha fusion protein.

Hinokiflavone treated cells accumulate hyper-SUMOylated proteins, which we hypothesise results from hinokiflavone inhibiting de-SUMOylation. Consistent with this, we show that in vitro

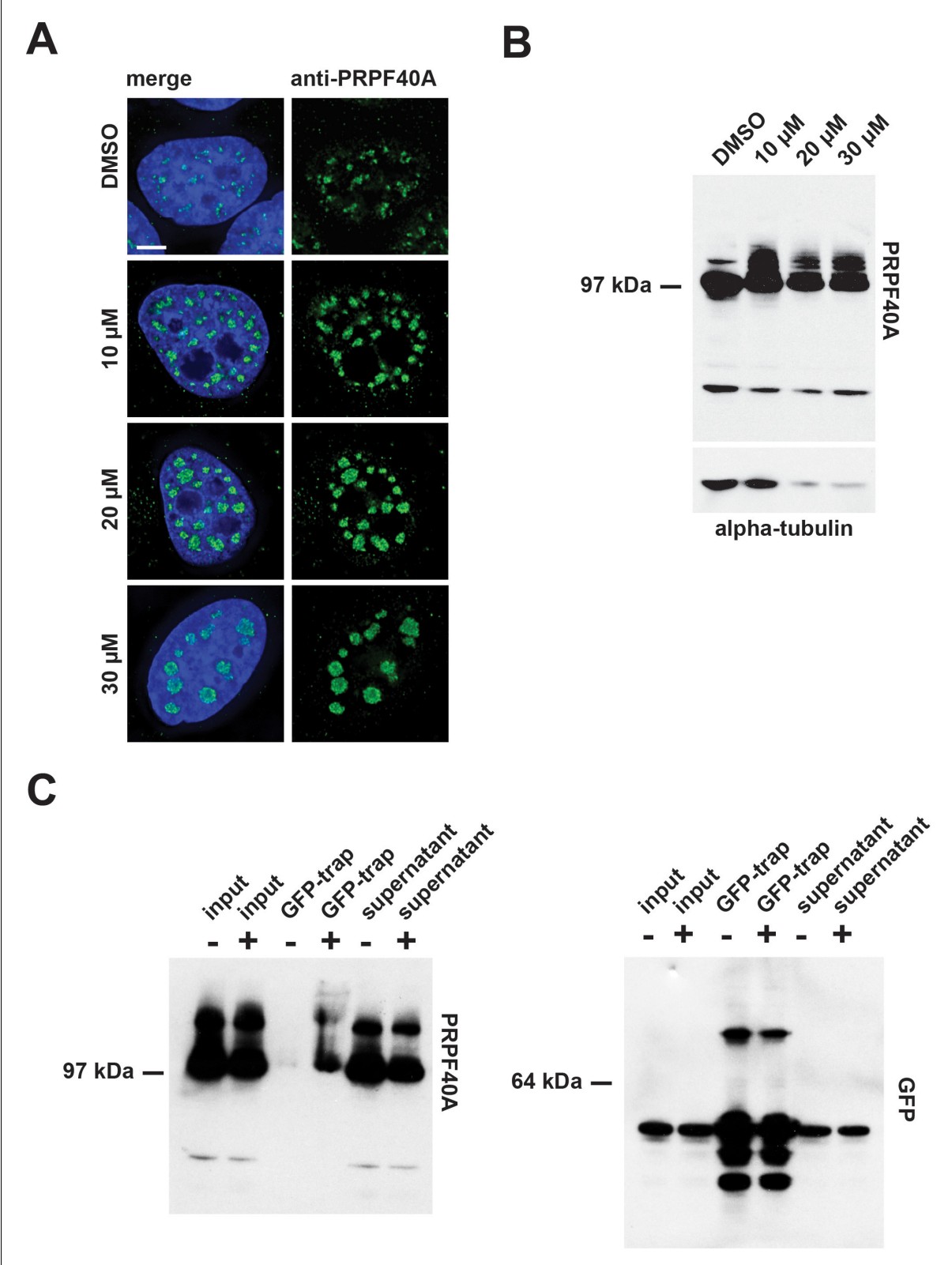

**Figure 15.** Confirmation of PRPF40A as a SUMO target protein. (**A**) Immunofluorescence analysis shows that hinokiflavone treatment leads to the relocation of PRPF40A to mega speckles in HeLa cells. Scale bar represents 6.5 µm. (**B**) HEK293 cells were treated with either DMSO (control), or with 10 µM, 20 µM, or 30 µM hinokiflavone, for 24 hr, then total cell lysate proteins were size-separated by SDS-PAGE, transferred to membranes, probed using the anti-PRPF40A antibody and visualized using chemiluminescence. In the presence of hinokiflavone additional higher molecular weight bands are

*Figure 15 continued on next page*

Figure 15 continued

detected. (C) YFP-SUMO2 expressing HeLa cells were treated either with DMSO (-), or 20 µM hinokiflavone (+), for 8 hr. Cells were lysed and the YFP-SUMOylated proteins immunoprecipitated with GFP-trap beads. The input, IPs, pellets and unbound fractions of both the control and hinokiflavone treated cells were size separated by SDS-PAGE, transferred to membranes and probed using anti-PRPF40A and anti-GFP antibodies and visualized using chemiluminescence.

DOI: https://doi.org/10.7554/eLife.27402.024

hinokiflavone can inhibit the catalytic activity of a purified, *E. coli* expressed fragment of the SUMO protease SENP1. Using a quantitative, mass spectrometry-based assay we also identified protein targets and mapped lysine residues showing increased levels of SUMO2 modification in hinokiflavone treated cells. The major hyperSUMOylated target proteins were enriched in pre-mRNA splicing factors, in particular, six components of the U2 snRNP spliceosome subunit, which is required for A complex formation. This included three lysine residues in the U2 snRNP protein PRPF40A, with K241, K375 and K517 showing a remarkable increase of >20 fold in levels of SUMO2 modification after hinokiflavone treatment. Our data thus provide multiple lines of evidence linking increased SUMO modification of proteins in the pre-mRNA splicing machinery with downstream effects on alternative splicing, nuclear organization and cell cycle progression.

To date, the best studied group of splicing modulators are the SF3B1 inhibitors, such as Spliceostatin A, which either have been isolated from the broth of bacteria, or else are synthetic derivatives of these compounds (*Kaida et al., 2007*; *Kotake et al., 2007*; *Webb et al., 2013*). Several other natural compounds isolated from plants have also been shown to affect pre-mRNA splicing, including the flavones apigenin (*Arango et al., 2013*) and luteolin (*Chiba et al., 2016*) and the biflavone isoginkgetin (*O'Brien et al., 2008*), but in comparison with the SF3B1 inhibitors, very little is known about either their targets, or their mode of action. A recent study reported that isoginkgetin may affect splicing by modulating RNA polymerase elongation rates (*Boswell et al., 2017*). As discussed below, we did not see evidence for hinokiflavone causing any major change in transcription. Furthermore, we observe that both isogingketin and hinokiflavone inhibit splicing of pre-synthesised transcripts in vitro, uncoupled from transcription. In this study, we compared the ability of the biflavones amentoflavone, cupressuflavone, hinokiflavone and sciadopitysin, to modify splicing, either in vitro, or *in cellulo*, comparing them with the previously described biflavone splicing inhibitor isoginkgetin. Our analysis showed that within this group of compounds hinokiflavone was the most potent biflavone modulator of splicing.

Hinokiflavone is a biapigenin found in many different plant families (*Harborne and Bryant, 1989*). Previous studies, using in silico screens, have suggested several different proteins as potential targets for hinokiflavone, including the prostaglandin D2 synthetase (*Fong et al., 2015*) and the matrix metalloproteinase-9 (*Kalva et al., 2014*). We also tested the ability of hinokiflavone to inhibit a large panel of purified kinases in vitro and observed that it had either no, or only weak, non-specific inhibitory effects on any of the 120 different kinases tested (data not shown). In contrast, however, using direct biochemical assays, we identify here that hinokiflavone is a SUMO protease inhibitor. Specifically, we show that, (a) hinokiflavone inhibits the catalytic activity of a highly purified, *E. coli* expressed fragment of the SUMO protease SENP1 and (b) that treatment of multiple human cell lines with hinokiflavone resulted in a dramatic increase in the levels of SUMO1 and SUMO2/3 modified proteins. The effect in hinokiflavone treated cells appears to be specific to SUMO modification because we detected little or no parallel increase in cells in the levels of proteins linked with related post-translational protein modifiers, that is, either ubiquitin, or NEDD8. Accumulation of high molecular weight SUMOylated proteins was also observed in vitro when HeLa nuclear extracts were incubated with hinokiflavone.

We characterized in more detail the stimulation of SUMO modification in cells treated with hinokiflavone, using a previously developed, quantitative, MS-based proteomics assay in HEK293 SUMO2^T90K cells (*Tammsalu et al., 2014*). We identified 924 SUMO2 modified lysine residues in 543 target proteins and showed that 22 of these lysine residues increased SUMO2 levels more than five-fold after hinokiflavone treatment. Interestingly, this unbiased assay independently linked the effect of hinokiflavone with the pre-mRNA splicing machinery, with ten of the lysine residues showing the highest increase in SUMO2 modification located specifically in six U2 snRNP proteins.

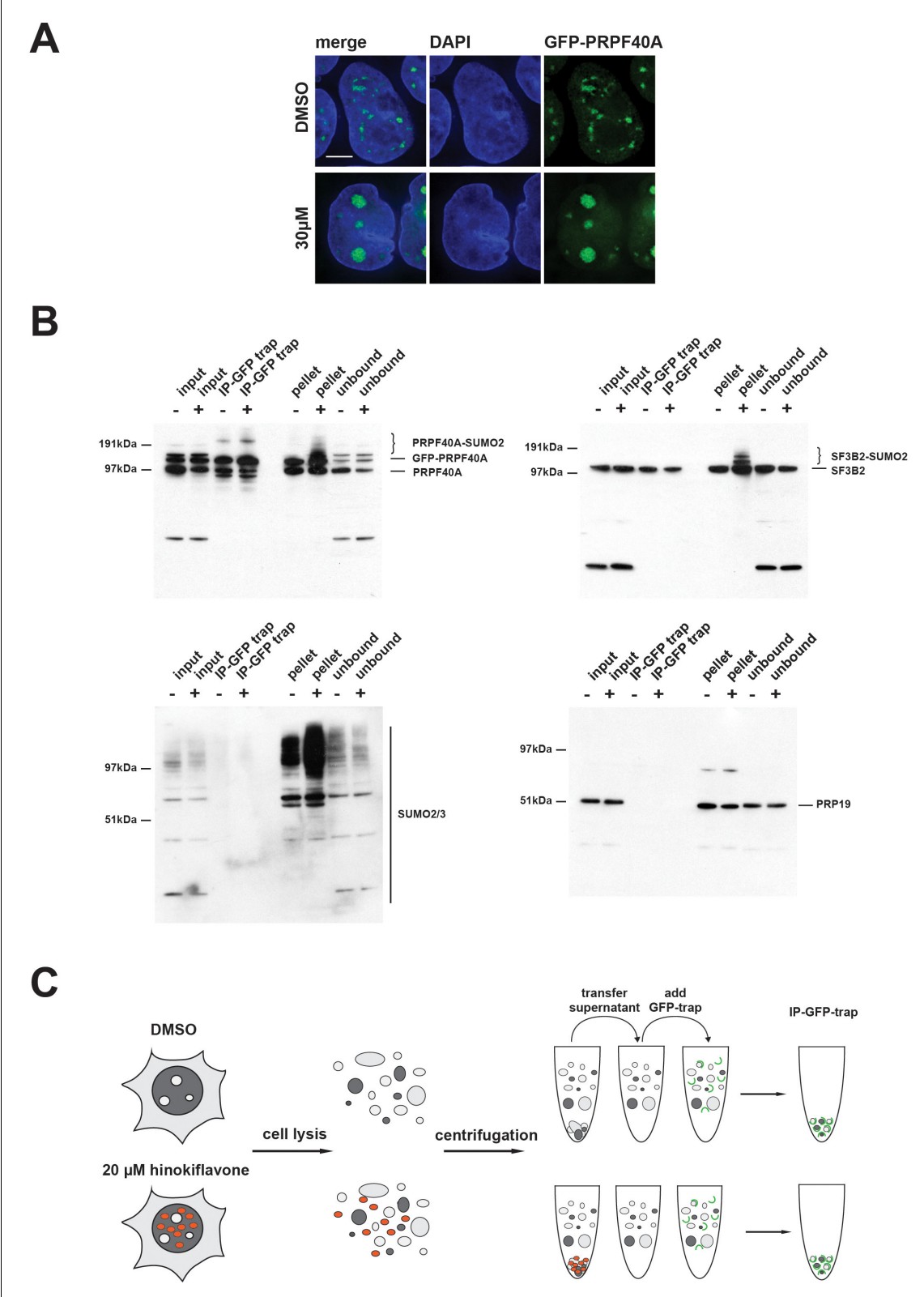

**Figure 16.** SUMOylated PRPF40A and SF3B2 accumulate in the insoluble fraction of HEK293 cell lysates. (**A**) GFP-PRPF40A stably expressed in HEK293 cells relocates to mega speckles in the presence of 30 μM hinokiflavone, as shown by fluorescence microscopy analysis. Scale bar represents 6.5 μM. (**B**) HEK293 cells stably expressing GFP-PRPF40A were treated with either DMSO (-) as a negative control, or with 20 μM hinokiflavone (+), for 8 hr. Cells were lysed and analyzed by co-immunoprecipitation (Co-IP). The input, IPs, pellets and unbound fractions of both the control and hinokiflavone treated

*Figure 16 continued on next page*

*Figure 16 continued*
cells were size separated by SDS-PAGE, transferred to membranes and probed using anti-PRPF40A, anti-SF3B2, anti-SUMO2/3 and anti-PRP19
antibodies and visualized using chemiluminescence. (C) Schematic representation of the Co-IP procedure.
DOI: https://doi.org/10.7554/eLife.27402.025

We note that U2 snRNP is required for assembly of the pre-splicing A complex and that hinokiflavone prevents spliceosome assembly in vitro proceeding beyond formation of the A complex. It is possible, therefore, that one or more U2 snRNP proteins are transiently SUMOylated during the spliceosome assembly cycle and must be de-SUMOylated for assembly to proceed beyond the A complex. For example, we speculate this could be involved in a proof reading mechanism that ensures accurate selection of 5′ and 3′ splice sites before proceeding to form a catalytically active complex. This would provide a potential mechanism linking our observations that hinokiflavone both inhibits protein de-SUMOylation and prevents spliceosome assembly in vitro proceeding beyond the A complex. This would be consistent with our additional observations by fluorescence microscopy that specifically splicing factors involved in the early steps of the splicing process are enriched in the megaspeckles that are formed in hinokiflavone treated cells, while other splicing factors associated with later steps in spliceosome assembly remain mostly diffusely distributed throughout the nucleus.

Our current data mapping lysine residues in U2 snRNP proteins that are SUMO2 modified and which show enhanced SUMO2 modification after hinokiflavone treatment, are consistent with previous data suggesting a potential link between splicing and protein SUMOylation. A recent study by Pozzi et al., showed that several splicing factors were SUMO2-modified at different stages of the splicing reaction in vitro, including several components of U2 snRNP (SF3B1, SF3B3, SF3A1, SF3A2, SF3A3). It was further shown that inhibition of PRPF3 SUMOylation prevented the interaction of the U4/U6 di-snRNP with U5 to form the tri-snRNP (*Pozzi et al., 2017*). Interestingly, we note that hinokiflavone also caused an increase in SUMO2 modification (>4 fold) on lysine 376 in PRPF3 (*Supplementary file 1*). Pelisch et al., reported that the Ser/Arg-rich non-snRNP protein splicing factor SRSF1 regulates protein SUMOylation and interacts with the SUMO E3 ligase PIAS1 (*Pelisch et al., 2010*). Interestingly, SRSF1 was originally identified independently as both an essential splicing factor in vitro and also as a factor involved in the control of alternative splice site choice in cells (*Cáceres and Krainer, 1993*; *Zuo and Manley, 1993*; *Xiao and Manley, 1998*). Detailed biochemical studies have shown that SRSF1 is multifunctional, with important roles during early steps in spliceosome assembly, leading to formation of the A complex.

Two recent proteomics studies that systematically identified proteins that are SUMO2 modified in cells, using high-throughput, mass spectrometry-based proteomics methods, reported splicing factors amongst the many targets detected in HEK293 (*Tammsalu et al., 2014*) and HeLa cells (*Hendriks et al., 2015*). Both these studies identified PRPF40A as a SUMO2 target protein. In addition, Tammsalu et al., identified SNRPD2 and Hendriks et al., identified the U2 snRNP components SF3A2, SF3B1 and SF3B2 as targets for SUMO2 modification. Furthermore, in a recent study identifying the SENP inhibitor SI2, it was reported that blocking SENP activity increased the levels of SUMOylation of multiple proteins, including the splicing factors USP39, SF3B1 and PRPF40A (*Wen et al., 2014*). In this latter study, the authors did not map the specific lysine residues that were modified by SUMO2 and none of these three studies investigate whether either SUMO modification, or inhibition of SENP activity, affects pre-mRNA splicing. It is nonetheless interesting that here we also find highly increased SUMO2 modification of the same U2 snRNP proteins, SF3B1 and PRPF40A, upon inhibition of SENP activity by hinokiflavone.

PRPF40A is a conserved spliceosome protein, present throughout eukaryotes from yeast to mammals. It was identified as a component of the U2 snRNP spliceosome subunit and suggested to have a role in mediating interactions between the separate 5′ and 3′ splice sites on pre-mRNAs (*Makarov et al., 2012*; *Abovich and Rosbash, 1997*). While other proteins may also influence the effects of hinokiflavone on spliceosome assembly, including potentially interactions between SUMO1, as well as SUMO2 modified targets (which we were not able to address here), it is nonetheless striking that all the splicing factors that we detect here as showing the highest increase (>five fold) in SUMO2 levels in cells treated with hinokiflavone, that is, PRPF40A, SF3A2, SF3B1, SF3B2, SNRPD1 and U2SURP, are either core components of, or associated with, U2 snRNPs and

have been shown to interact with each other (*Makarov et al., 2012*). Several reversible, post-translational modifications, including phosphorylation, ubiquitination, methylation and acetylation, were previously reported to be important for the assembly and the disassembly of the spliceosome (*Chen and Moore, 2014*). Our present data suggest that reversible SUMO modification may also be an important feature of mechanisms affecting spliceosome assembly and alternative splice site choice.

Previous studies using the SF3B1-targeted splicing inhibitors have shown that they are potent growth inhibitors of many different cancer cell lines (*Sakai et al., 2002*; *Asai et al., 2007*). When used in combination with BCL-2 inhibitors, these SF3B1 inhibitors have been demonstrated to induce apoptosis in small lung cancer cells, chronic lymphocytic leukemia cells and head and neck cancer cells (*Gao and Koide, 2013*; *Larrayoz et al., 2016*; *Gao et al., 2014*). This has been mainly attributed to their ability to change the alternative splicing of MCL1 pre-mRNAs. Indeed, meayamycin and spliceostatin A showed the greatest effect on exon two skipping towards the pro-apoptotic product of MCL1, when compared with their effects on alternative splicing of 34 other genes important for cell proliferation and apoptosis (*Papasaikas et al., 2015*). Here we show that, in common with the previously demonstrated effect of these SF3B1 inhibitors, hinokiflavone treatment also changes the alternative splicing of MCL1 pre-mRNA to favour the production of the proapoptotic isoform MCL1-S, over the antiapoptotic isoform MCL1-L. These differential effects of splicing inhibitors on alternative splicing likely reflect the differential requirements for splicing factors at different introns within a transcript.

The effect of hinokiflavone on MCL1 splicing and its ability to induce cell cycle arrest and/or apoptosis, raise the possibility that either hinokiflavone itself, or a synthetic derivative thereof, could be developed in future as a novel cancer therapeutic. We observed significant variation in the dose response to hinokiflavone between human cancer cell lines. The greatest effect on cell survival was seen in the human acute promyeolytic leukemia cell line NB4, where extensive apoptosis was induced upon treatment with 10 μM hinokiflavone. This positively correlated with the large effect of hinokiflavone in also altering alternative pre-mRNA splicing in NB4 cells, including promoting splicing of the pro-apoptotic isoform of MCL1. However, it is plausible that an increase in PML SUMOylation also contributes to the extreme sensitivity of NB4 cells to hinokiflavone. For example, our MS data in HEK293 cells showed ~twofold increased SUMO2 modification of PML after hinokiflavone treatment. It has been shown that the use of arsenic ($As_2O_3$) for therapeutic treatment of acute promyelocytic leukemia (APL) patients is effective because it triggers degradation of PML and oncogenic PML fusion proteins, such as PML-RARalpha, through promoting their hyper-SUMOylation (*Ferhi et al., 2016*). We propose that hinokiflavone could have a similar effect through its ability to block SENP activity and therefore stimulate accumulation of hyper-SUMOylated proteins.

In summary, this study has identified the plant derived biflavone hinokiflavone as an inhibitor of SUMO protease activity that affects spliceosome assembly and splice site selection, leading to major changes in alternative splicing patterns in human cancer cell lines. Our development of an efficient route for the chemical synthesis of hinokiflavone (King et al., unpublished), has allowed us to confirm that the hinokiflavone molecule is the active compound and will facilitate the future investigation of its detailed mechanism of action and exploration of its development as a potential cancer therapeutic. This should help to clarify the relation between SUMO modification and alternative splicing and allow a better understanding of how hinokiflavone exerts such a potent increase in SUMOylation of U2 snRNP proteins.

## Materials and methods

### Compounds

Hinokiflavone, amentoflavone, cupressuflavone and sciadopitysin were purchased from Extrasynthese (Genay Cedex, France) and isoginkgetin was purchased from MerckMillipore (Darmstadt, Germany).

### Cell culture, RNA isolation and RT-PCR

HeLa (RRID:CVCL_0030), HEK293 (RRID:CVCL_0045) and NB4 (RRID:CVCL_0005) cells were purchased from ATCC and cultured in Dulbecco's modified Eagle's medium supplemented with 10%

fetal bovine serum, 2 mM glutamine (Life Technologies, Carlsbad, CA, USA) and 100 µg/ml strepto-mycin (100X stock, Life Technologies). Total RNA was extracted from cells using the NucleoSpin RNA II Kit (Macherey-Nagel, Düren, Germany), according to the manufacturer's instructions. 200 ng of total RNA was reverse transcribed and amplified using the One Step RT-PCR kit (QIAGEN, Hilden, Germany), according to the manufacturer's instructions. Primer sequences are listed in *Table 2*.

## In vitro transcription and RT-based in vitro splicing reaction

The Adenovirus pre-mRNA Ad1 and HPV18 E6 pre-mRNA were in vitro transcribed using the RNA-MAxx High Yield Transcription Kit (Agilent, Santa Clara, CA, USA), according to the manufacturer's instructions, followed by a DNAase 1 digestion and RNA clean up using the RNeasy Kit (QIAGEN, Hilden, Germany).

Standard splicing reactions were carried out in 30% HeLa nuclear extract (Computer Cell Culture Centre, Seneffe, Belgium), in the presence of either DMSO, or compound, and incubated at 30 °C for 90 min. The splicing reaction was followed by a heat inactivation step of 5 min at 95 °C before the samples were subjected to proteinase K (QIAGEN, Hilden, Germany) digestion for 30 min at 55 °C and another heat inactivation step at 95 °C for 5 min. The spliced and non-spliced RNA was ampli-fied using the One step RT-PCR Kit (QIAGEN, Hilden, Germany), according to the manufacturer's instructions. PCR products were separated by electrophoresis using 1% agarose gels containing SYBR safe DNA gel stain (Life Technologies, Carlsbad, CA, USA).

## Radioactive in vitro splicing reaction and native gels

Radioactive in vitro splicing reactions were performed as previously described (*Lamond et al., 1987*) using a $^{32}$P labelled pBsAd1 pre-mRNA substrate.

Splicing complexes were analysed on a low melting point agarose gel (1.5%, w/v), as previously described (*Konarska and Sharp, 1987*) and visualized by phosphor imaging (Typhoon 8600, GE Healthcare, Pittsburgh, PA, USA).

## Gel-based SENP activity assay

SENP activity assays were carried out as 20 µl reactions containing 2 µl 10x reaction buffer (200 mM Hepes, 500 mM NaCl, 30 mM MgCl$_2$, pH 7.5), recombinant, *E.coli* expressed SENP1 fragment com-prising amino acids 415–643 (186 nM) in SENP buffer (50 mM Tris-HCl, 150 mM NaCl) and 5 µM SUMOylated template YFP-RANGAP (aa 418–587)-ECFP-SUMO2 (5 µM). Either 1 µL DMSO alone (control), or 1 µL compound dissolved in DMSO was added and the reactions were incubated at 37°C for 15 min then stopped by adding 5 µL 4x LDS loading buffer. After incubating the samples at 70°C for 10 min the proteins were separated on a 4–12% Tris-Bis PAGE gel and visualized with Coo-massie blue.

## DARTS assay

To recombinant *E.coli* expressed SENP1 fragment comprising amino acids 415–643 (186 nM) in 20 µl SENP buffer (50 mM Tris-HCl, 150 mM NaCl), either 1 µL DMSO (control), or 1 µL compound dis-solved in DMSO, was added and the reactions were incubated at 4°C for 1 hr. The samples were then treated with different concentrations of pronase (Sigma-Aldrich, St. Louise, MO, US) for 30 min at RT. The reactions were stopped by adding 5 µL 4x LDS loading buffer. After incubating the sam-ples at 70°C for 10 min the proteins were separated by SDS PAGE and visualized with Coomassie blue.

## Cellular thermal shift assay

To 1 ml of HeLa NE, either 50 µl of DMSO, or 50 µl of 10 mM hinokiflavone, was added and incu-bated at RT for 20 min. DMSO as well as hinokiflavone treated NE samples were then split into seven 100 µl aliquots. Each sample was incubated at a specific temperature (30°C, 33°C, 37°C, 40°C, 43°C, 47°C or 50°C) for 3 min, which was followed by 3 min incubation at RT. After ultracentrifuga-tion at 35,000 rpm, 4°C for 20 min using a Optima MAX ultracentrifuge and a TLA 120.2 rotor, the supernatant was transferred to a new tube containing 25 µl 4x LDS buffer and boiled for 10 min at 95°C. The samples were subjected to western blot analysis with the indicated primary antibodies.

**Table 2.** Primary antibodies used for IF staining and/or western-blotting.

| Primary antibody | Company |
| --- | --- |
| anti-BCAS2 | Abcam, Cambridge, UK (RRID:AB_1861326) |
| anti-CDC5L | BD Transduction Laboratories (RRID:AB_399724) |
| anti-coilin | Proteintech, Chicago, IL, US (RRID:AB_2276345) |
| anti-CSTF2 | Santa Cruz, US (RRID:AB_668179) |
| anti-CTNNBL1 | Abcam, Cambridge, UK (RRID:AB_1523420) |
| anti-DDX46 | Proteintech, Chicago, IL, US (RRID:AB_2090927) |
| anti-Fibrillarin | Cytoskeleton Inc, Denver, CO, US (RRID:AB_10709399) |
| anti-PRPF40A | Novus, Biologicals, Chambridge, UK (RRID:AB_11012473) |
| anti-PLRG1 | Abcam, Cambridge, UK (RRID:AB_2170868) |
| anti-PRP19 | Abcam, Cambridge, UK (RRID:AB_2170868) |
| anti-NEDD8 | Cell Signaling Technology, Danvers, MA, US (RRID:AB_659972) |
| anti-SF3B1 | Abcam, Cambridge, UK (RRID:AB_2186512) |
| anti-SF3B2 | Novus Biologicals, Abington, UK (RRID:AB_1110397) |
| anti-SFRS1 | Abcam, Cambridge, UK (RRID:AB_298608) |
| anti-SMN | ImmunoGlobe, Himmelstadt, Germany (RRID:AB_2687973) |
| anti-SNRPA1 | Abcam, Cambridge, UK (RRID:AB_11139816) |
| anti-snRNP200 | Abcam, Cambridge, UK (RRID:AB_10901078) |
| anti-SR proteins | Merck Group, Darmstadt, Germany (RRID:AB_10807429) |
| anti-SUMO1 | Cell Signaling Technology, Danvers, US (RRID:AB_10698887) |
| anti-SUMO2/3 | Cell Signaling Technology, Danvers, US (RRID:AB_2198425) |
| anti-TMG | Merck Group, Darmstadt, Germany (RRID:AB_2687977) |
| anti-U1A | Iain Mattaj (RRID:AB_2713922) |
| anti-U2AF65 | SIGMA, ST Louis, Missouri, US (RRID:AB_262122) |
| anti-Ubiquitin | Cell Signaling Technology, Danvers, US (RRID:AB_2180538) |
| anti-Y12 | Joan Steitz (RRID:AB_2692320) |

DOI: https://doi.org/10.7554/eLife.27402.026

## Cell fixation and immunofluorescence analysis

HeLa, HEK293 and NB4 cells were each treated either with DMSO alone (control), or with compound dissolved in DMSO, then grown on cover slips in DMEM for either 4 hr, 8 hr or 24 hr, at 37°C before fixing with 4% paraformaldehyde in PHEM buffer for 10 min at RT. After rinsing the cells with PBS, the cells were permeabilized with 0.5% Triton X100 in PBS prior to incubation with the primary antibodies (see *Table 3*). After incubation with the primary antibody for 1 hr at RT, the cover slips were washed twice with 0.5% Tween-20 in PBS for 5 min before they were incubated with the dye-conjugated secondary antibody for 30 min. Cells were then stained with DAPI (Sigma-Aldrich, St. Louis, MO, USA) and the cover slips were mounted in Vectashield medium (Vector Laboratories, Peterborough, UK). The samples were visualized using a fluorescence microscope (Zeiss, Jena, Germany; Axiovert-DeltaVision Image Restoration; Applied Precision, LLC).

## Fluorescence in situ hybridisation

HeLa cells were treated either with DMSO alone (control), or with compound dissolved in DMSO, then grown on cover slips in DMEM for either 4 hr, 8 hr or 24 hr, at 37°C before fixing with 4% paraformaldehyde in PHEM buffer for 10 min at RT. Incubation of the cells in ice cold methanol for 10 min was followed by an incubation in 70% ethanol for 15 min. The cells were then washed once with Tris-HCl pH 8.0 before the hybridization buffer [yeast tRNA (1 mg/ml), 0,005% BSA, 10% dextran sulphate, 25% deionized formamide], containing 5'- labelled Cy3 Oligo-dT (30) at a final concentration of 1 ng/μl was added to the cells and incubated overnight at 37°C. After the hybridization, the cells were washed once with 4x SSC and twice with 2x SSC. Cells were then stained with DAPI (Sigma-Aldrich, St. Louis, MO, USA) in 2x SSC, 0.1% Triton-X-100 for 5 min and the cover slips were mounted in Vectashield medium (Vector Laboratories, Peterborough, UK). The samples were visualized using a fluorescence microscope (Zeiss, Jena, Germany; Axiovert-DeltaVision Image Restoration; Applied Precision, LLC).

## Flow cytometry analysis

Cells were seeded in 12 well plates and after 24 hr were treated with either DMSO alone (control), or compound dissolved in DMSO. The cells were grown at 37 °C then harvested after either 4 hr, 8 hr, or 24 hr, then washed twice with PBS, resuspended in cold 70% ethanol and fixed at RT for 30 min. Fixed cells were then washed twice with PBS and resuspended in PI stain solution (50 μg/ml

**Table 3.** Primer pairs used for RT-PCR to identify pre-mRNA splicing changes.

| Primer | Sequence |
| --- | --- |
| ACTR1b for | CCGCTCAACCCGAGTAAGAA |
| ACTR1b rev | CAGCCGAGGTATGGAAGTCA |
| DXO for | TGGGGAGGTTAACACCAACG |
| DXO rev | GCTCTGGGAAAGCTAAGGA |
| EIF4A2 for | GTCTCTCCTTCGTGGCATCT |
| EIF4A2 rev | TCTCCCGGGTGTACCAACA |
| HSP40 for | GAACCAAAATCACTTTCCCCAAGGAAGG |
| HSP40 rev | AATGAGGTCCCCACGTTTCTCGGGTGT |
| MCL1 for | GAGGAGGAGGAGGACGAGTT |
| MCL1 rev | ACCAGCTCCTACTCCAGCAA |
| NOP56 for | GCATCCACAGTGCAGATCCT |
| NOP56 rev | GCAATCGATTCGTGAGGCAA |
| FAS for | CCCGGCCCAGAAATACCAAG |
| FAS rev | GACTCCAGCAATAGTGGTGATA |
| RIOK3 for | GCTGAAGGACCATTTATTACTGGAG |
| RIOK3 rev | TTCTTGCTGTGTTCTTTCTCCCACAC |

DOI: https://doi.org/10.7554/eLife.27402.027

propidium iodide and 100 µg/ml ribonuclease A in PBS). Cells were incubated in PI stain solution for 30 min and then analyzed by flow cytometry on a BD FACScalibur. The flow cytometry data were analyzed using FlowJo (TreeStar Inc, Ashland, OR, USA).

## Pulse labeling of HeLa cells with 5-ethynyluridine (EU)

Newly synthesized RNA was detected by using the Click-iT RNA imaging Kit (Life Technologies, Carlsbad, US). In brief, HeLa cells were grown at 37°C for 4 or 8 hrs in the presence of either DMSO (control), or compound dissolved in DMSO, then pulse labeled for 20 min with 5-ethynyluridine (EU). The cells were then fixed and the Click-iT detection was performed, according to the manufacturer's instructions.

## Western blot analysis

Cells treated as described were harvested, washed with PBS and lysed in 1x LDS buffer (Life Technologies, Carlsbad, US). Proteins were separated using a 4–12% NuPAGE Bis-Tris gel (Life, Technologies, Carlsbad, US) and transferred to a nitrocellulose blotting membrane (Amersham, Little Chalfont, UK) by electroblotting. Target proteins were detected with the help of the WesternBreeze Chemiluminescent Kit, according to the manufacturer's instructions.

## Cell culture conditions and protein extraction

For quantitative SUMO2 modification site-specific proteomic experiments, the adherent HEK293[6His-SUMO2-T90K] N3S cells were cultured in DMEM lacking L-lysine, L-arginine and L-glutamine (Biosera, Uckfield, UK) and supplemented with 10% dialysed FBS, 1 µg/ml puromycin, 100 U/ml Pen-Strep, 2 mM L-glutamine, and either natural [($^{12}C_6$, $^{14}N_2$; K0) and ($^{12}C_6$, $^{14}N_4$; R0)] (Sigma-Aldrich, St. Louise, MO, US), or heavy stable isotope-containing [($^{13}C_6$, $^{15}N_2$; K8) and CNLM-539-H ($^{13}C_6$, $^{15}N_4$; R10)] (Cambridge Isotope Laboratories, Tweksbury, US) 0.8 mM L-lysine and 0.14 mM L-arginine. Five 175 cm$^2$ dishes of either light-, or heavy-labeled cells at ~90% confluency were transferred into SMEM lacking L-lysine, L-arginine and L-glutamine (Biosera, Uckfield, UK) and supplemented with 10% dialysed FBS, 1 µg/ml puromycin, 100 U/ml Pen-Strep, 2 mM L-glutamine, and either natural, or heavy stable isotope-containing 0.8 mM L-lysine and 0.14 mM L-arginine, respectively. Approximately 1.0 L of suspension culture, corresponding to ~4.7 $\times$ 10$^8$ cells, was cultured per experimental condition and heavy-labelled cells were treated with 20 µM hinokiflavone in DMSO for 8 hr, whereas an equivalent volume of DMSO alone was added to the light-labelled control culture. Cells were harvested by centrifugation, washed with cold 1 $\times$ Dulbecco's PBS (DPBS) and 0.9 g of each cell pellet was mixed. The combined pellet of cells was lysed in fresh cell lysis buffer containing 6 M guanidine hydrochloride, 100 mM sodium phosphate buffer pH 8.0, 10 mM Tris-HCl pH 8.0, 10 mM imidazole and 5 mM 2-mercaptoethanol. Lysis buffer was added in a ratio of 25:1 (vol/wt). DNA was disrupted by short pulses of sonication, insoluble material was removed by centrifugation and filtration through sterile Minisart NML syringe filters with 0.2 µm pore size (Sartorius, Epsom, UK) and protein concentration was estimated using Pierce bicinchoninic acid (BCA) protein assay (Thermo Fisher Scientific, Waltham, US).

## Nickel affinity chromatography and protein digestion

Approximately 98.5% of cell lysate protein was used for the identification of SUMO2 modification sites and peptides were prepared essentially as previously described by (*Tammsalu et al., 2015*). Cell lysate protein was mixed with pre-equilibrated Ni$^{2+}$-NTA agarose resin (QIAGEN, Hilden, Germany) in a ratio of 200:1 (wt/vol) and the slurry was rotated overnight at 4°C. Beads were collected into an empty spin chromatography column (Bio-Rad, Hercules, UK) and washed with five resin volumes of cell lysis buffer, ten resin volumes of wash buffer pH 8.0 (8 M urea, 100 mM sodium phosphate buffer pH 8.0, 10 mM Tris-HCl pH 8.0, 10 mM imidazole, 5 mM 2-mercaptoethanol), ten resin volumes of wash buffer pH 6.3 (8 M urea, 100 mM sodium phosphate buffer pH 6.3, 10 mM Tris-HCl pH 8.0, 10 mM imidazole, 5 mM 2-mercaptoethanol) and ten resin volumes of wash buffer pH 8.0. Conjugates were eluted in three sequential steps with two resin volumes of elution buffer (200 mM imidazole in wash buffer pH 8.0) and protein concentration was estimated by UV-visible spectrophotometry at 280 nm. Protein digestion was performed on an ultrafiltration spin column with a 30 kDa nominal molecular weight cutoff limit (Sartorius, Epsom, UK). Denatured proteins were concentrated

on the device, washed twice with 8 M urea, 100 mM Tris-HCl pH 7.5 and treated with 50 mM 2-chloroacetamide in 8 M urea, 100 mM Tris-HCl pH 7.5 at room temperature for 20 min in the dark. Samples were washed once with 8 M urea, 100 mM Tris-HCl pH 7.5, twice with IAP buffer (50 mM MOPS-NaOH pH 7.2, 10 mM Na$_2$HPO$_4$, 50 mM NaCl) and digested overnight with Lysyl endopeptidase (Lys-C; Wako, Neuss, Germany) in IAP buffer at 37°C [enzyme-to-protein ratio 1:50 (wt/wt)]. Peptides were collected and the device was washed with IAP buffer to increase the yield of Lys-C digested peptides. High-molecular-weight peptides retained on the 30 kDa ultrafiltration spin column were subsequently digested with endoproteinase Glu-C (Sigma-Aldrich, St Louis, US) overnight in IAP buffer at 25°C [enzyme-to-protein ratio 1:100 (wt/wt)] and after the collection of peptides, the device was washed with IAP buffer.

## Immunoaffinity purification of diGly-Lys-containing peptides

Peptides in IAP buffer were supplemented with ~18.75 µg of K-ε-GG-specific antibody (Cell Signalling Technology, Danvers, MA, US) BS3 cross-linked to 3 µl of protein A agarose resin and rotated overnight at 4°C. Agarose beads were washed twice with 50 resin volumes of cold IAP buffer and peptides were eluted in two sequential steps with 50 µl of 0.15% (vol/vol) trifluoroacetic acid (TFA).

## In solution digestion of cell lysate proteins

Complete proteome analysis was performed with ~120 µg of cell lysate protein dissolved in 6 M urea, 2 M thiourea after TCA-based precipitation. The proteins were treated with 1 mM DTT for 1 hr and 5 mM 2-chloroacetamide for 1.5 hr at room temperature in the dark, prior to four-fold dilution into 50 mM ammonium bicarbonate and digestion with Lys-C at enzyme-to-protein ratio 1:50 (wt/wt) at room temperature for 4.5 hr. Lys-C digested samples were then divided in two, and one of the samples was diluted twofold into 50 mM ammonium bicarbonate followed by supplementation with trypsin at enzyme-to-protein ratio 1:50 (wt/wt). Both samples were digested overnight at room temperature. Resulting peptide mixtures were fractionated into six fractions based on the pH of the solution (pH 11.0, 8.0, 6.0, 5.0, 4.0, and 3.0) used to elute the peptides from a pipette tip-based Empore anion exchanger (Agilent Technologies, Santa Clara, US) according to a protocol described in *Wiśniewski et al. (2009)*.

## Liquid chromatography (LC)-tandem mass spectrometry (MS/MS)

Prior to MS-based analyses, all peptide samples were desalted on self-made Empore C18 (Agilent Technologies, cat. no. 12145004) Stop and Go Extraction Tips (StageTips) according to a protocol by *Rappsilber et al. (2007)*. Desalted peptide samples were analysed using EASY-nLC 1000 nanoflow UHPLC system, EASY-Spray ion source and Q Exactive hybrid quadrupole-Orbitrap mass spectrometer (all Thermo Fisher Scientific). Peptides were loaded onto 2 cm Acclaim PepMap 100 C18 nanoViper pre-column (75 µm ID; 3 µm particles; 100 Å pore size) at a constant pressure of 800 bar and separated using 50 cm EASY-Spray PepMap RSLC C18 analytical column (75 µm ID; 2 µm particles; 100 Å pore size) maintained at 45°C.

DiGly-Lys-enriched samples were analyzed at least twice. Exploratory analysis using standard MS settings was performed using 10% of the sample and peptides were separated with 60 min linear gradient of 5–22% (vol/vol) acetonitrile in 0.1% (vol/vol) formic acid at a flow rate of 250 nl/min, followed by a 12 min linear increase of acetonitrile to 40% (vol/vol). Total length of the gradient including column washout and re-equilibration was 90 min. Comprehensive analyses of diGly Lys-containing peptide samples were performed using identical LC conditions. Peptides corresponding to the complete human proteome were analysed with 240 min linear gradient of 5–22% (vol/vol) acetonitrile in 0.1% (vol/vol) formic acid at a flow rate of 250 nl/min with the majority of peptides eluting during a 220 min acetonitrile window from 5% to 50% (vol/vol).

Peptides eluting from the liquid chromatography column were charged using electrospray ionisation and MS data were acquired online in a profile spectrum data format. Full MS scan covered a mass range of mass-to-charge ratio (m/z) either 300–1800, or 300–1600, during standard and comprehensive peptide analyses, respectively. Target value was set to 1,000,000 ions with a maximum injection time (IT) of 20 ms and full MS was acquired at a mass resolution of 70,000 at m/z 200. Data dependent MS/MS scan was initiated if the intensity of a mass peak reached a minimum of 20,000 ions. During standard LC-MS/MS analyses, up to 10 most abundant ions (Top 10) were selected

using 2 Th mass isolation range when centered at the parent ion of interest. For comprehensive analyses, up to 4 (Top 4) most abundant ions were picked for MS/MS. Selection of molecules with peptide-like isotopic distribution was preferred. Target value for MS/MS scan was set to 500,000 ions with a maximum IT of 60 ms and resolution of 17,500 at either m/z 200 for standard, or maximum IT of 500 ms and a resolution 35,000 at m/z 200 for comprehensive peptide analyses. Precursor ions were fragmented by higher energy collisional dissociation (HCD), using normalised collision energy of 30 and fixed first mass was set to m/z 100. Precursor ions with either undetermined, single, or high (>8) charge state were rejected. Ions triggering a data-dependent MS/MS scan were placed on the dynamic exclusion list for either 40 s (standard analyses), or 60 s (comprehensive analyses) and isotope exclusion was enabled. The sample of Lys-C digested diGly-Lys-containing peptides was analyzed twice according to the settings of the comprehensive analysis, however the 1126 peptide ions identified with the first comprehensive LC-MS/MS analysis were added to the exclusion list.

## Analysis of raw MS files

Raw mass spectrometric data files were processed with MaxQuant software (version 1.3.0.5) (*Cox and Mann, 2008*) and peak lists were searched with an integrated Andromeda search engine (*Cox et al., 2011*) against an entire human UniprotKB proteome containing canonical and isoform sequences (*UniProt Consortium, 2015*) downloaded in April 2013 and supplemented with the primary sequence of 6His-SUMO2$^{T90K}$. Raw files were divided into parameter groups based on the specificity of the proteolysis applied during sample preparation. Hydrolysis of peptide bonds C-terminal to either Lys, or Lys and Arg, with a maximum of three missed cleavages was allowed for peptides processed with either Lys-C, or with Lys-C and Trypsin, respectively. Samples acquired after an additional Glu-C digestion were analyzed with enzyme specificity set to C-terminal Lys, Glu and Asp with a maximum of five missed cleavages. Carbamidomethylation of cysteine residues was specified as a fixed modification and oxidation of methionines, acetylation of protein N-termini and where applicable, Gly-Gly adduct on internal lysine residues and phosphorylation of Ser, Tyr and Thr, were selected as variable modifications. Additional analyses were performed by including either deamidation of Gln and Asn, or conversion of N-terminal Glu or Gln to pyroglutamate as extra variable modifications. Maximum peptide mass of 10,000 Da was allowed, multiplicity was set to 2, and K8 and R10 were selected as heavy-labelled counterparts. A maximum of either six, or four, labelled residues were allowed per peptide during the analyses of either diGly-Lys-enriched samples, or peptides representing the entire proteome, respectively. A decoy sequence database was generated using Lys as a special amino acid when analysing diGly-Lys-containing peptides. Default values were chosen for the rest of the parameters. All datasets were filtered by posterior error probability to achieve a false discovery rate of 1% at protein, peptide and modified site level. Peptides were identified using a reverse decoy database search and potential contaminants of non-human origin were excluded. The list of SUMO2 modification sites (*Supplementary file 1*) is based on diGly-Lys containing peptides, where modification site localization probability is greater than 75% and mass error is less than 2 ppm, or 4 ppm in the case of unsuccessful recalibration. The non-normalized SILAC ratios of diGly-Lys-containing peptides were normalized according to the median normalization factor obtained for the complete proteome experiment and was based on the non-normalized and normalized protein ratios reported in the MaxQuant output file. Raw MS data available from PRIDE repository, accession number PXD007629.

## Stable cell lines

cDNA encoding PRPF40A was purchased from GenScript (Piscataway, US) and cloned into the pEGFP-C3 vector using EcoRI/BamHI restriction sites. HEK293 cell lines stably expressing GFP-PRPF40A were generated by transfecting the expression constructs using Effectene (QIAGEN, Hilden, Germany), according to the manufacturer's instructions and maintaining the cells in DMEM (low glucose), with 10% FBS and 0.5 mg/ml G418 (Invitrogen, Carlsbad, USA).

## Immunoprecipitation assays

HEK293 cells stably expressing GFP-PRPF40A were seeded into 10 cm dishes for 24 hr before the cells were treated either with DMSO alone (control), or with 20 μM hinokiflavone in DMSO, for 8 hr before cells were trypsinised and harvested. Cells were washed twice with ice-cold PBS

before ~$1\times10^7$ cells were lysed in 1 ml of Co-IP buffer (1 mM EDTA, 100 µM Na$_3$VO$_4$, 0.5% Triton X-100, 20 mM beta-Glycerol P), with protease inhibitors and NBE for 30 min. Before adding 50 µl of GFP-Trap magnetic beads (Chromotek, Apple Vally, US) to 1 ml of cleared lysate, the lysed cells were centrifuged for 10 min at 13,000 rpm at 4°C. After incubation at 4°C overnight, the beads were washed three times with Co-IP lysis buffer and proteins were eluted from the beads by adding 200 µl 1x LDS buffer and boiling the samples for 10 min at 95°C. The samples were subjected to western blot analysis with the indicated primary antibodies.

HeLa cells stably expressing YFP-SUMO2 were seeded into 10 cm dishes for 24 hr before the cells were treated either with DMSO alone (control), or 20 µM hinokiflavone in DMSO, for 8 hr before cells were trypsinised and harvested. The cells were then lysed in RIPA buffer [50 mM Tris–HCl (pH 7.5), 1% (v/v) NP-40, 0.5% (w/v sodiumdeoxycholate, 0.05% (w/v) SDS, 1 mM EDTA, 150 mM NaCl] containing protease inhibitors for 15 min and the lysates were homogenized using QIAshredders. 50 µl of GFP-Trap magnetic beads (Chromotek, Apple Valley, US) were added to 1 ml of cleared lysate that had been centrifuged for 10 min at 13,000 rpm at 4°C. After incubation with lysate at 4°C overnight, the beads were washed three times with RIPA buffer and proteins were eluted from the beads by adding 150 µl 1x LDS buffer and boiling the samples for 10 min at 95°C. The samples were subjected to western blot analysis with the indicated primary antibodies.

## Acknowledgements

We thank our colleagues for advice and discussion. We thank staff in the Fingerprints, Flow Cytometry and Cell Sorting Facility for advice and assistance (CAST, University of Dundee).

## Additional information

### Funding

| Funder | Grant reference number | Author |
| --- | --- | --- |
| Wellcome | 073980/Z/03/B | Angus I Lamond |
| Wellcome | 097045/B/11/Z | Angus I Lamond |
| Wellcome | 098391/Z/12/7 | Ronald T Hay |
| Wellcome | 105606/Z/14/Z | Ronald T Hay |
| European Commission | PITN-GA-2011-290257 | Triin Tammsalu |
| Engineering and Physical Sciences Research Council | EP/L50497X/1 | Lewis J King Helmi Kreinin |

The funders had no role in study design, data collection and interpretation, or the decision to submit the work for publication.

### Author contributions

Andrea Pawellek, Conceptualization, Investigation, Visualization, Methodology, Writing—original draft, Writing—review and editing; Ursula Ryder, Investigation, Methodology; Triin Tammsalu, Data curation, Investigation, Methodology, Writing—review and editing; Lewis J King, Helmi Kreinin, Resources, Investigation, Methodology; Tony Ly, Data curation, Methodology, Writing—review and editing; Ronald T Hay, Resources, Supervision, Funding acquisition, Writing—review and editing; Richard C Hartley, Supervision, Funding acquisition, Writing—review and editing; Angus I Lamond, Conceptualization, Supervision, Funding acquisition, Writing—review and editing

### Author ORCIDs

Angus I Lamond http://orcid.org/0000-0001-6204-6045

### Decision letter and Author response

Decision letter https://doi.org/10.7554/eLife.27402.032
Author response https://doi.org/10.7554/eLife.27402.033

## Additional files

### Supplementary files

• Supplementary file 1. List of lysine residues identified to show an increase in SUMO2-modification in HEK293 cells 8 hr after treatment with 20 µM hinokiflavone. Corresponding gene and protein names are indicated for each lysine.
DOI: https://doi.org/10.7554/eLife.27402.028

• Transparent reporting form
DOI: https://doi.org/10.7554/eLife.27402.029

### Major datasets

The following dataset was generated:

| Author(s) | Year | Dataset title | Dataset URL | Database, license, and accessibility information |
|---|---|---|---|---|
| Pawellek A, Ryder U, Tammsalu T, King LJ, Kreinin H, Ly T, Hay RT, Richard C Hartley, Angus I Lamond | 2017 | Characterisation of the biflavonoid hinokiflavone as a pre-mRNA splicing modulator that inhibits SENP | https://www.ebi.ac.uk/pride/archive/projects/PXD007629 | Publicly available at PRIDE (accession no. PXD007629) |

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
