## [Decision Letter]

Thank you for submitting your article "Characterisation of the biflavonoid hinokiflavone as a pre-mRNA splicing modulator that inhibits SENP" for consideration by *eLife*. Your article has been favorably evaluated by James Manley (Senior Editor) and three reviewers, one of whom, Juan Valcárcel, is a member of our Board of Reviewing Editors.

The reviewers have discussed the reviews with one another and the Reviewing Editor has drafted this decision to help you prepare a revised submission.

Summary:

In this manuscript Pawellek et al. describe their identification of the plant biflavonoid hinokiflavone as a modulator of pre-mRNA splicing in vitro and of alternative pre-mRNA splicing in mammalian cells. Moreover, they show that in vitro hinokiflavone inhibits pre-mRNA splicing by stalling spliceosome assembly at the A complex stage and thus blocking B complex formation. IF studies of cells treated with hinokiflavone revealed enlarged speckles and relocalisations of splicing factors to the speckles that are known to be involved in A complex formation. On the basis of the localisation of SUMO1 and 2 in the enlarged speckles, the authors then show that hinokiflavone increases protein SUMOylation, and biochemical assays indicate that the drug inhibits SENP1 SUMO protease. By quantitative MS and immunoblotting studies they finally show increased levels of SUMO2 modification in hinokiflavone-treated cells. Strikingly, several proteins associated with U2 snRNP were found to be particularly enriched among SUMO2-modified proteins and decreased interaction between some of these factors. The authors therefore conclude that increased SUMOylation of U2 proteins in the presence of hinokiflavone may lead to mis-regulation/inhibition of pre-mRNA splicing.

This is a solid and comprehensive analysis and a welcome addition to the rather limited toolbox of small molecule modulators of pre-mRNA splicing. The study also provides a novel focused set of reversible post-translational modifications that act during the spliceosome cycle. Moreover, the manuscript will trigger further work to clarify the effects of these compounds and their eventual use for splicing modulation.

Essential modifications:

1) In view of the relatively moderate SUMOylation level of U2 proteins upon treatment of cells with hinokiflavone (as shown by western blots; Figure 16), it is striking that the inhibitor completely blocks B complex formation in vitro. Therefore, it is important for the authors to investigate the level of SUMOylation of the various relevant U2 proteins in the stalled A complexes. This could be accomplished by immunoblotting using purified stalled A complexes. While the correlation between increased SUMOylation of several U2 proteins and the observed stalling of A complexes in the presence of hinokiflavine is highly suggestive, the authors cannot at present exclude the possibility that additional (non-A-complex) spliceosomal proteins may also be hyperSUMOylated in the presence of the drug, which may also contribute to splicing inhibition.

2) Also related to Figure 16: the authors should make an effort to better explain in the text how the experiment was conducted and also label the panels in ways that will facilitate a more direct interpretation of the result (e.g. labeling the positions expected for the different protein species and their sumoylated forms; it might also help to provide a small schematic to show the initial separation of insoluble vs. insoluble, followed by the co-IP of the soluble). The western blot panels also need to be annotated to indicate the main bands corresponding to each protein, as well as the SUMOylated species. Are the size markers correct for SF3B2, which is 145 kDa? It might help to state explicitly that the initial hypothesis (that PRPF40A SUMOylation affects interactions with other proteins) could not be addressed due to the presence of SUMOylated protein only in the insoluble pellet. In addition, the authors should explain better their interpretation of this result: on one hand, PRPF40A and SF3B2 interact under normal conditions, on the other, their sumoylated forms are only present in insoluble aggregates; it is therefore unclear if sumoylation directly affects their interaction or the modification simply makes these proteins – and perhaps other proteins – insoluble and therefore part of an aggregate. How does formation of an aggregate contribute to splicing regulation? Simply by decreasing the overall levels of these proteins / U2 snRNP?

3) Some of the figures would greatly benefit from data quantification. Specifically:

a) Subsection “Hinokiflavone blocks cell cycle progression” and Figure 4. Axes should be labelled clearly (or explained in the legend). It would be better to use the same scale for all panels, allowing a fairer comparison between DMSO and hinokiflavone. The claims made in the text are not all obviously supported by the data in Figure 4.g. "dependent on time and concentration" but only one time point and concentration are shown. It's presumably relevant to the statement about apoptosis that the main peak in the NB4 + hinikoflavone panel is to the left of the main peaks in all other panels? Perhaps aligning the two panels for each cell line vertically would be better to facilitate comparison – or superimpose the two sets of data in different colours? Overall, the text needs to provide a clearer justification for the conclusions stated, possibly with additional data shown or minimally reference to "data not shown" if necessary.

b) Figure 5. The relocalization to "megaspeckles" is clear for SRSF2, U1A and U2AF65. It's much less clear for DDX46, SART1 and pan-SR (at least from the images shown) – the SR speckles actually appear larger in DMSO than hinikoflavone. It would be good to provide some kind of quantitative analysis of size and/or number of speckles/megaspeckles per cell, and to define a threshold for "megaspeckle" size.

c) Figure 13. It would be good to see some quantification of the protease sensitivity with an indication of reproducibility (e.g. expressed as% full length compared to no digestion). Alternatively, some additional biophysical support for direct interaction of hinokiflavone and SENP1 would be good (e.g. thermal melt, as used for fragment screening in drug discovery).

d) Figure 15 and text: "…showed that the level of SUMOylated PRPF40A increased…". Strictly speaking the data show accumulation of larger species of PRPF40A, consistent with SUMOylation. Proof that these are SUMOylated species would require combined IP and western blots.

e) Figure 15. Quantitative analysis of speckle size and/or number per cell is needed for this panel – the differences are not so obvious.

4) The authors should discuss their findings with respect to a recent report by Srebrow's group (Pozzi et al. NAR 2017) on the role of protein SUMOylation in modulating pre-mRNA splicing. While this report suggested the requirement of sumoylation for accumulation or recruitment of the U5/4/6 tri-snRNP to form complex B, the results of Pawellek et al. argue for a role of de-sumoylation of U2 snRNP components in the transition from A to B.

---

## [Author Response]

Essential modifications:1) In view of the relatively moderate SUMOylation level of U2 proteins upon treatment of cells with hinokiflavone (as shown by western blots; Figure 16), it is striking that the inhibitor completely blocks B complex formation in vitro. Therefore, it is important for the authors to investigate the level of SUMOylation of the various relevant U2 proteins in the stalled A complexes. This could be accomplished by immunoblotting using purified stalled A complexes. While the correlation between increased SUMOylation of several U2 proteins and the observed stalling of A complexes in the presence of hinokiflavine is highly suggestive, the authors cannot at present exclude the possibility that additional (non-A-complex) spliceosomal proteins may also be hyperSUMOylated in the presence of the drug, which may also contribute to splicing inhibition.

We agree that in principle it cannot be excluded that other proteins are also hyperSUMOylated that affect the observed splicing inhibition, apart from the proteins we identify that are known to be involved in A complex formation. However, we point out that this was exactly why we carried out as part of our study an in depth, unbiased proteomics screen using a state of the art, quantitative MS approach (Tammsalu et al., 2014). We have therefore included in this manuscript data that comprehensively map the SUMO-modified lysine residues in target proteins that are affected by hinokiflavone. We have described this SUMO2 target screen in detail in the manuscript. We feel that the results showing that 6 of the most highly SUMO2 modified proteins detected were found to be U2 snRNP proteins is striking, as is the unprecedented level of increased SUMO2 modification we detected (~145 fold!) on the PRPF40A protein. We have now added an additional IP experiment in the revised manuscript, using a stable cell line expressing FP-tagged SUMO2, which shows that in this cell background hinokiflavone treatment for 8hrs results in SUMOylation of >15% of the PRPF40A protein. We respectfully feel that further mechanistic studies on how the different protein targets of SUMO2 affect spliceosome assembly and alternative splicing will have to be addressed and published in separate future in depth analyses, particularly since the observed effects on splicing may depend upon an interplay between SUMO2 and SUMO1 modified proteins, which we cannot currently address (this possibility is now mentioned in the revised manuscript). In the revised manuscript we have tried to present a very balanced picture of what our data show and how this relates to other studies in the literature on the effects of SUMO modification on pre-mRNA splicing.

2) Also related to Figure 16: the authors should make an effort to better explain in the text how the experiment was conducted and also label the panels in ways that will facilitate a more direct interpretation of the result (e.g. labeling the positions expected for the different protein species and their sumoylated forms; it might also help to provide a small schematic to show the initial separation of insoluble vs. insoluble, followed by the co-IP of the soluble).

We have revised the manuscript to address this. We now indicate the SUMOylated bands on the blots in panel B and have added a schematic as requested (new Figure 16, panel C).

The western blot panels also need to be annotated to indicate the main bands corresponding to each protein, as well as the SUMOylated species. Are the size markers correct for SF3B2, which is 145 kDa?

We have annotated the bands in the blots shown in Figure 16. We are confident that the size markers run on this gel system as indicated and we confirmed that the commercial antibody we used to detect SF3B2 is specific by siRNA knock-down (not shown).

It might help to state explicitly that the initial hypothesis (that PRPF40A SUMOylation affects interactions with other proteins) could not be addressed due to the presence of SUMOylated protein only in the insoluble pellet.

We agree and have now made this explicit statement in the penultimate paragraph of the Results section.

In addition, the authors should explain better their interpretation of this result: on one hand, PRPF40A and SF3B2 interact under normal conditions, on the other, their sumoylated forms are only present in insoluble aggregates; it is therefore unclear if sumoylation directly affects their interaction or the modification simply makes these proteins – and perhaps other proteins – insoluble and therefore part of an aggregate. How does formation of an aggregate contribute to splicing regulation? Simply by decreasing the overall levels of these proteins / U2 snRNP?

Our interpretation of the data is that extensive protein hyperSUMOylation causes aggregates to form when cell lysates are made. SUMO is known to promote protein-protein interactions and complex formation in cells and our thinking here is informed by the expertise of our collaborator Ron Hay, who is a world leader in studying the biochemistry of SUMOylation and its effects on protein function. We have revised the manuscript to indicate this.

3) Some of the figures would greatly benefit from data quantification. Specifically:a) Subsection “Hinokiflavone blocks cell cycle progression” and Figure 4. Axes should be labelled clearly (or explained in the legend). It would be better to use the same scale for all panels, allowing a fairer comparison between DMSO and hinokiflavone. The claims made in the text are not all obviously supported by the data in Figure 4.g. "dependent on time and concentration" but only one time point and concentration are shown. It's presumably relevant to the statement about apoptosis that the main peak in the NB4 + hinikoflavone panel is to the left of the main peaks in all other panels? Perhaps aligning the two panels for each cell line vertically would be better to facilitate comparison – or superimpose the two sets of data in different colours? Overall, the text needs to provide a clearer justification for the conclusions stated, possibly with additional data shown or minimally reference to "data not shown" if necessary.

We have included a new revised version of Figure 4, which we hope illustrates clearly that different cell lines respond differently to hinokiflavone and that each cell line shows a concentration-dependent effect for a given time of treatment (24 hrs in the revised figure). We show in other figures also that time of exposure to hinokiflavone affects the cellular response (e.g. Figure 5—figure supplement 1).

b) Figure 5. The relocalization to "megaspeckles" is clear for SRSF2, U1A and U2AF65. It's much less clear for DDX46, SART1 and pan-SR (at least from the images shown) – the SR speckles actually appear larger in DMSO than hinikoflavone. It would be good to provide some kind of quantitative analysis of size and/or number of speckles/megaspeckles per cell, and to define a threshold for "megaspeckle" size.

Our statements in the manuscript of course reflected our overall analysis of a large number of separate experiments and cell stainings etc. in the course of this study. To ensure that the data shown in Figure 5 better reflect our conclusions, based upon data from all of these analyses, we have prepared a new version of this figure, using several new examples of fluorescence images that illustrate the increase in accumulation of the proteins in mega speckles. In addition, we have included additional details, including counting the numbers of mega speckles and their respective sizes in different cells, for a representative experiment (data for this is illustrated in new Figure 5—figure supplement 1). We have also included new experiments, which showed that the mega speckles also contain polyadenylated RNA (new Figure 8) and we have added reference to this in the Abstract, Results and Discussion sections.

c) Figure 13. It would be good to see some quantification of the protease sensitivity with an indication of reproducibility (e.g. expressed as% full length compared to no digestion). Alternatively, some additional biophysical support for direct interaction of hinokiflavone and SENP1 would be good (e.g. thermal melt, as used for fragment screening in drug discovery).

To address this we have performed additional thermal profiling analyses comparing the stability of endogenous SENP1 in HeLa nuclear extracts at different temperatures, either in the presence, or absence, of hinokiflavone. These data are included in the new Figure 13 panel C. These data strongly support our conclusion that hinokiflavone binds directly to SENP1, as discussed in the manuscript.

d) Figure 15 and text: "…showed that the level of SUMOylated PRPF40A increased…". Strictly speaking the data show accumulation of larger species of PRPF40A, consistent with SUMOylation. Proof that these are SUMOylated species would require combined IP and western blots.

We agree and therefore performed the additional IP and blotting experiment suggested, which is now shown in new Figure 15, panel C. This confirms that indeed levels of SUMOylated PRPF40A increase following hinokiflavone treatment, consistent with the MS data summarised in Figure 14.

e) Figure 15. Quantitative analysis of speckle size and/or number per cell is needed for this panel – the differences are not so obvious.

We have replaced Figure 15 panel A with a new micrograph, which shows the effect of increased speckle size more clearly.

4) The authors should discuss their findings with respect to a recent report by Srebrow's group (Pozzi et al. NAR 2017) on the role of protein SUMOylation in modulating pre-mRNA splicing. While this report suggested the requirement of sumoylation for accumulation or recruitment of the U5/4/6 tri-snRNP to form complex B, the results of Pawellek et al. argue for a role of de-sumoylation of U2 snRNP components in the transition from A to B.

We have included reference to and discussion of this work in the revised manuscript. We note that hinokiflavone also caused an increase in SUMO2 modification (>4-fold) on lysine 376 in PRPF3.